# LEARNING ANTI-CLASSES WITH ONE-COLD CROSS ENTROPY LOSS

## ABSTRACT

While softmax cross entropy loss is the standard objective for supervised classification, it primarily focuses on the ground truth classes, ignoring the relationships between the non-target, complementary classes. This leaves valuable information unexploited during optimization. In this work, we set explicit non-zero target distributions for the complementary classes, in order to address this limitation. Specifically, for each class, we define an *anti-class*, which consists of everything that is not part of the target class—this includes all complementary classes as well as out-of-distribution samples, and in general any instance that does not belong to the true class. Various distributions can be used as a target for the anti-classes. For example, by setting a uniform one-cold encoded distribution over the complementary classes as a target for each anti-class, we encourage the model to equally distribute activations across all non-target classes. This approach promotes a symmetric geometric structure of classes in the final feature space, increases the degree of neural collapse during training, addresses the independence deficit problem of neural networks and improves generalization. Our extensive evaluation demonstrates that our proposed framework consistently results in performance gains across multiple settings, including classification, open-set recognition, and out-of-distribution detection.

## 1 INTRODUCTION

Deep learning has seen amazing progress over the past decade across multiple domains such as vision, speech, and text (He et al., 2015b). The optimization criterion universally used for supervised classification tasks is the cross entropy loss, which aims at minimizing the discrepancy between the predicted probability distribution across classes and the ground truth probability distribution, typically represented as a one-hot encoding of the true label (Goodfellow et al., 2016). Minimizing the cross entropy loss maximizes the likelihood of correctly identifying disjoint classes under a given set of parameters of a statistical learning model, such as a neural network, making it an intuitive choice, with strong empirical results over the years.

Despite its theoretical and practical appeal, cross entropy loss is not without its limitations. Neural networks optimized with cross entropy are known to result in overconfident predictions (Guo et al., 2017), (Nguyen et al., 2015). This problem becomes more evident in open set classification scenarios, where a model must be able to also recognize unknown (i.e., not seen in training) classes at inference time (Scheirer et al., 2013). Current neural networks struggle to recognize out-of-distribution (OOD) samples (Bendale & Boult, 2015), and often misclassify them to one of the known classes with high confidence, which calls into question their trustworthiness. In a recent work, Feng et al. (2024) also identified an independence deficit in neural networks, where the prediction confidence of some classes is redundantly determined by others through simple linear relationships, even when the classes are semantically unrelated. This entanglement can lead to overfitting and poor generalization, as the network neglects learning distinct class representations.

In spite of these limitations, neural networks can generalize remarkably well, and understanding the underlying reasons has been the focus of extensive research (Zhang et al., 2021), (Yang et al., 2020). In this effort, recent work has uncovered the intriguing empirical phenomenon of Neural Collapse (NC), which arises during the terminal stage of deep neural network training (Papyan et al., 2020), (Zhu et al., 2024). In NC, class features collapse to their respective means, class means become

maximally separated in a Simplex Equiangular Tight Frame (ETF), and the classifier aligns with these class means, leading to a highly structured decision boundary. NC has been linked to improved robustness and benign overfitting (Bartlett et al., 2020), (Ma et al., 2018), with connections to the information bottleneck theory (Tishby & Zaslavsky, 2015), which suggests that networks focus on essential information, while discarding unnecessary variability.

While independence deficit and neural collapse are distinct and even contradicting behaviors—one reflecting a redundancy in class independence and the other signifying maximally separated and independent class representations— cross entropy loss does not provide a mechanism to explicitly control these phenomena. Instead, their emergence during training is often incidental, with networks passively exhibiting one behavior or the other, depending on the dataset size, class distribution, task difficulty and model complexity rather than actively optimizing for either. In this work, we rethink the classification objective by introducing the concept of an *anti-class*, which consists of the complementary set of a class, encompassing all that the class is not. With anti-classes, we can explicitly control the relationships between the complementary classes, and therefore the structure of the classes in the final feature manifold.

We propose using a one-cold encoding, where the true class is labeled with 0, while the rest of the classes are labeled with non-zero targets. Inspired by the geometric properties of the Simplex ETF, we mainly focus on a uniform one-cold encoding, where each complementary class is labeled with 1, as illustrated in Figure 1a. This encoding reflects that all complementary classes of a ground truth class belong equally to its anti-class. (see Figure 1b). Through this simple label transformation, each class is encouraged to maintain a symmetric relationship with all other classes, similar to the symmetry in the Simplex ETF, where class centroids are equinorm and equiangular. This approach forces the network to develop distinct and independent representations for each class, preventing reliance on linear correlations with other classes. We argue that this enhances the model's generalization capability.

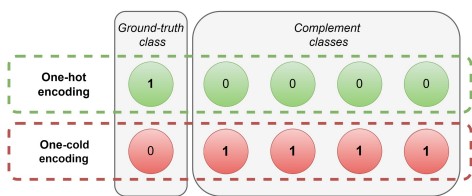 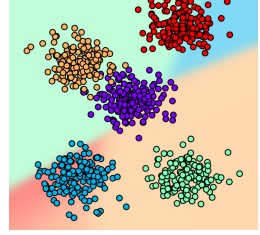 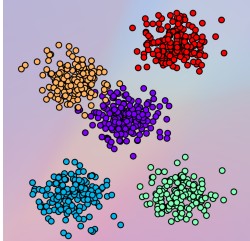

(a) One-hot encoding vs one-cold encoding.

(b) Anti-class regions of one-hot cross entropy (left) and one-cold cross entropy (right).

Figure 1: (b) Each point in the grid corresponds to a specific $(x_1, x_2)$ coordinate in the input space $\mathcal{X}$, (not limited to visible class samples). Each point is passed through a trained MLP, its logits are negated and passed through a softmax to provide an anti-class distribution (membership of that specific point of the grid to each anti-class). The color assigned to each point is calculated with the linear combination between the anti-class distribution of that point and the colors of the classes. For example, green and yellow classes become negatively correlated, even if these classes are semantically unrelated. This can result in spurious correlations and independence deficit. In contrast, uniform OCCE removes these connections by enforcing a uniform anti-class distribution, where all complementary classes are equally represented in each anti-class.

Our key contributions are highlighted as follows:

- We introduce the concept of anti-classes, providing a novel approach to modeling classification problems by directly optimizing the relationships between complementary classes.
- We propose using one-cold cross entropy (OCCE) to enable the model to learn target anti-class distributions, in either a shared or decoupled layer from the class distributions, allowing to control the geometric structure of classes.
- We analyze the impact of the uniform OCCE objective on neural collapse, independence deficit, and the generalization capabilities of neural networks.
- Through extensive experiments, we demonstrate that our proposed method consistently improves performance in both closed-set and open-set classification tasks.

## 2 METHOD

### 2.1 ANTI-CLASS FORMULATION

Let's consider a predefined set of $N$ known classes. Each class $k$ can be represented as a subspace in a $d$-dimensional space, denoted by $\mathcal{C}_k \subseteq \mathbb{R}^d$. For every class $k$, we can define the complementary subspace to $\mathcal{C}_k$, which we denote as $\bar{\mathcal{C}}_k = \mathbb{R}^d \setminus \mathcal{C}_k$ and refer to as the anti-class subspace of $k$. The anti-class subspace $\bar{\mathcal{C}}_k$ contains every instance that does not belong to class $k$, including instances from other known or unknown classes, out-of-distribution instances, random noise or adversarial examples. Intuitively, directly modeling the anti-class subspace $\bar{\mathcal{C}}_k$ seems a much harder problem than modeling $\mathcal{C}_k$, as the latter is clearly more restricted. Moreover, in mutually exclusive classes, an instance that belongs to $\mathcal{C}_k$ inherently belongs to the anti-class of all other known classes $\bar{\mathcal{C}}_{j \neq k}$. Consequently, in multi-class classification every vector $\mathbf{p} \in \mathbb{R}^d$ belongs to at least $N - 1$ anti-classes, and potentially to all $N$ of them, if the vector does not represent an instance of a known class. We can encode the anti-class labels using a one-cold encoded label $\bar{\mathbf{l}}$, where the ground truth class is labeled with 0 (hence the term cold), and every complement class is labeled with a non-zero value, indicating its membership to the anti-class. If we adopt hard assignments, the vector $\bar{\mathbf{l}}$ can be defined as:

$$\bar{l}_i = \begin{cases} 0 & \text{if } i = \text{ground truth class,} \\ 1 & \text{if } i = \text{complementary class.} \end{cases} \tag{1}$$

### 2.2 ORDINARY SUPERVISED CLASSIFICATION

In a standard supervised learning scenario, we deal with a set of $n$ labeled instances $\mathcal{D} = \{(\mathbf{x}_0, l_0), \ldots, (\mathbf{x}_n, l_n)\}$ over an initial feature space $\mathcal{X} \subseteq \mathbb{R}^d$ and a discrete label space $L = \{1, \ldots, N\}$, where $d$ is the dimensionality of $\mathcal{X}$ and $N$ the number of classes. An instance $\mathbf{x}$ which belongs to class $k$ is sampled from the subspace $\mathcal{C}_k$. Given a model with parameters $\theta$, the objective of an ordinary classification task is to find the mapping $f : \mathcal{X} \to L$, where $f(\mathbf{x}; \theta) = l$, correctly mapping each input $\mathbf{x}$ to its corresponding label $l$. Standard classification training minimizes the cross entropy loss for each instance, comparing the model's predicted probability distribution $\hat{\mathbf{y}}$ with the one-hot encoded label $\mathbf{y}$. The cross entropy loss is defined as:

$$L_{\text{CE}} = -\mathbf{y}^\top \log(\hat{\mathbf{y}}), \quad \text{where} \quad \hat{y}_i = \frac{\exp(z_i)}{\sum_{j=1}^{C} \exp(z_j)}. \tag{2}$$

In equation 2, only the log-probability of the ground truth labels is explicitly minimized, while the probabilities of the complementary classes are indirectly influenced through the softmax normalization. This approach overlooks the relationships between the complementary classes. Such a limitation can be problematic, particularly for handling anomalous samples like adversarial examples or out-of-distribution inputs. Ideally, for these cases, a model should produce a close to uniform probability distribution across all classes, indicating maximum uncertainty. However, achieving this uniform distribution requires equal activations $\mathbf{z}$ for all classes, which is not explicitly encouraged by cross entropy loss. Furthermore, it does not enforce any specific geometric structure of the classes in the feature space, leading to undesirable phenomena such as minority collapse (Fang et al., 2021), where in imbalanced training the features of minority classes collapse towards identical clusters, or independence deficit (Feng et al., 2024), where spurious correlations between classes emerge.

### 2.3 LEARNING WITH ANTI-CLASSES

**Deriving the OCCE Loss.** To address these challenges, we shift our focus to anti-classes. Since one-cold encoded labels are not valid probability distributions, standard cross entropy loss cannot be directly applied. A trivial solution would be to convert the task into a multi-label problem and predict each anti-class probability independently using binary cross entropy loss. However, this approach is redundant, as binary cross entropy treats labels 0 and 1 identically, learning each class independently without controlling the relationships between complementary classes. To resolve this, we introduce competition between anti-classes by applying a softmax function. While in one-hot encoding, this results in a winner-takes-all scenario (Goodfellow et al., 2016), with anti-classes, the target probability distribution can reflect how much an instance belongs to each anti-class. In this

work, we mainly focus on a uniform one-cold encoding (hard assignments on anti-class labels $\bar{\mathbf{l}}$), where a sample fully belongs to all anti-classes except its true class, implying that all anti-classes are treated equally as winners. The target anti-class probability distribution $\bar{\mathbf{y}}$ can be simply defined for each instance as:

$$\bar{y}_i = \frac{\bar{l}_i}{\sum_{j=1}^{N} \bar{l}_j} = \begin{cases} 0 & \text{if } i = \text{ground truth class,} \\ \frac{1}{N-1} & \text{if } i = \text{complementary class.} \end{cases} \tag{3}$$

Let now $\bar{\mathbf{z}}$ be the output activation vector of our network. For a network optimized to learn anti-classes, higher activations correspond to higher probability of not belonging to the corresponding class. We can now define the one-cold cross entropy loss as:

$$L_{\text{OCCE}} = -\bar{\mathbf{y}}^\top \log(\hat{\bar{\mathbf{y}}}), \quad \text{where} \quad \hat{\bar{y}}_i = \frac{\exp(\bar{z}_i)}{\sum_{j=1}^{N} \exp(\bar{z}_j)} \tag{4}$$

Finally, to obtain class activations $\mathbf{z}$ we negate the anti-class activations $\mathbf{z} = -\bar{\mathbf{z}}$ before softmax.

OCCE can be used also with non-uniform target distributions, as we demonstrate in the supplementary material (see A.1.1), to handle different problems, e.g., accounting for different inter-class similarities. When a uniform one-cold target distribution is used as a standalone objective, then we acquire, as a special case, the reverse cross entropy (RCE) loss (Pang et al., 2017), which has been proposed as a substitute of cross entropy loss for adversarial robustness. However, the RCE loss encounters challenges with training stability on larger datasets. On the other hand, the proposed approach, can effectively overcome these challenges by combining class and anti-class learning. This combination results in consistent improvements across many classification tasks, distinguishing our approach from RCE, which, as empirically shown, is not competitive in these scenarios.

**Convergence Analysis.** The gradients returned by one-hot cross entropy (CE) and by uniform one-cold cross entropy (OCCE) lead to different convergence dynamics. The gradients with respect to the activations $\mathbf{z}$ under CE and uniform OCCE are:

$$\frac{\partial L_{CE}}{\partial z_i} = \begin{cases} \hat{y}_i - 1 & \text{if } i = c, \\ \hat{y}_i & \text{if } i \neq c, \end{cases} \quad \text{and} \quad \frac{\partial L_{OCCE}}{\partial \bar{z}_i} = \begin{cases} \hat{\bar{y}}_i & \text{if } i = c, \\ \hat{\bar{y}}_i - \frac{1}{N-1} & \text{if } i \neq c, \end{cases} \quad \text{respectively.} \tag{5}$$

When the gradient of CE loss with respect to the activations vector $\mathbf{z}$ approaches zero, $\nabla_{\mathbf{z}} L_{CE} \to \mathbf{0}$, and under no further constraints, we get $\hat{y}_c \to 1$, $z_c \to +\infty$ for the correct class c, implying that its predicted probability converges to 1, and its corresponding activation $z_c$ tends to positive infinity. Similarly $\hat{y}_k \to 0$, $z_k \to -\infty$ for any complementary class $k \neq c$, implying that the predicted probabilities $\hat{y}_k$ and activations $z_k$ of any complementary class independently approach 0 and negative infinity, respectively. In the case of uniform OCCE loss, when the class activations are the negated anti-class activations $\mathbf{z} = -\bar{\mathbf{z}}$ and the gradient satisfies $\nabla_{\bar{\mathbf{z}}} L_{OCCE} \to \mathbf{0}$, the asymptotic behavior with respect to the class activations $\mathbf{z}$ changes as follows:

$$\hat{\bar{y}}_c \to 0, \quad z_c \to +\infty \quad \text{for the correct class } c, \tag{6}$$

$$\hat{\bar{y}}_k \to \frac{1}{N-1}, \quad z_k \to -\log\left(\frac{\sum_{j \neq c} \exp(-z_j)}{N-1}\right) = z_{const} \quad \text{for any complementary class } k. \tag{7}$$

The limits in Equations 6 and 7 suggest that while the correct class activation $z_c$ still tends to positive infinity, the activations of the complementary classes tend to equal values, since the sum over complementary classes in the numerator is the same for all of them. Consequently, $z_{k_1} \to z_{k_2}$ for any $k_1, k_2 \neq c$.

**Gradients Behaviour.** In Figure 2, we analyze the gradient dynamics for both CE and OCCE under the same setup: a 3-class scenario where the activation for the correct class is fixed at $z_0 = 2.5$ and the activations for the two complementary classes, $z_1$ and $z_2$, vary within the range $[-5, 5]$. In Figure 2a, we visualize the gradient fields for both cases. Under CE, the correct class is encouraged to have a higher activation than the complementary classes, which are independently driven towards lower values. In contrast, under OCCE, the gradients for the correct class become significant only when both complementary classes have high activations. In the case of uniform OCCE, if the activations of the complementary classes have different values, the gradients work to equalize them by reducing the higher activation and increasing the lower one.

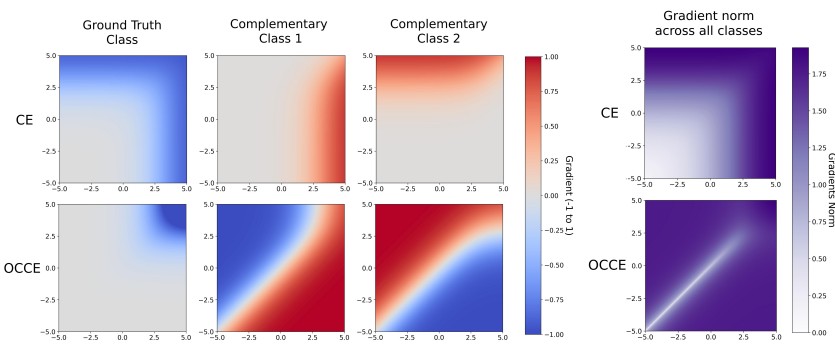

(a) Gradients for all classes                    (b) Gradients norm

Figure 2: Given a 3-class classification scenario, we fix the ground truth class activation $z_0 = 2.5$ and plot activations for complement classes $z_1$ and $z_2$ ranging from -5 to 5. (a) Each grid shows the gradients for each class at any $(z_1, z_2)$ point. (b) The cumulative gradient norm over all classes is shown.

In Figure 2b, we compare the cumulative gradient norms across all classes. For CE, large gradients appear when either complementary class has a competitive activation, without any regard for the relationship between the complementary classes. Consequently, for a classification task with N classes, gradients have small magnitude as long as $\max_{j \neq c} z_j \ll z_c$. For OCCE, high magnitude gradients are observed in most regions except where $\forall j \neq c : z_j = z_{\text{const}} \ll z_c$, which is a much more restricted area, than that of CE.

Although theoretical analyses show that neural networks trained with CE loss converge to a Simplex ETF under ideal optimization conditions (Lu & Steinerberger, 2022), in practice, the extent of neural collapse observed varies. This variation is influenced by factors such as the number of classes, their label distribution, the available data, and the complexity of the model. Empirically, neural collapse tends to occur only in the later stages of training when CE loss approaches zero, implying that the activations for all complementary classes converge towards zero, achieving nearly equal values. With uniform OCCE, this behavior can be explicitly enforced from earlier stages of training, leading to faster convergence toward this desired structure.

## 3 EXPERIMENTS

### 3.1 OCCE BEHAVIOUR ANALYSIS

**Unified CE+OCCE Objective.** The most straightforward way to apply an OCCE loss is on the negated class activations ($\bar{\mathbf{z}} = -\mathbf{z}$), as a unified objective with standard cross entropy loss applied on $\mathbf{z}$, as shown in Figure 3. However, OCCE can also be combined with any other loss that emphasizes the correct class. We also introduce a scalar $\gamma$ in order to control the contribution of OCCE in the final unified objective:

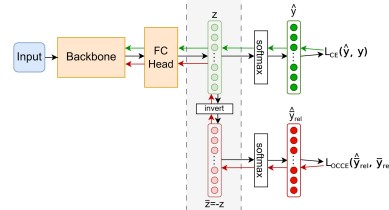

$$L = L_{\text{CE}} + \gamma \cdot L_{\text{OCCE}} \qquad (8)$$

Figure 3: Unified architecture.

**Neural Collapse.** To study the effect of uniform OCCE loss on the occurrence of neural collapse during optimization, we measure the amount of neural collapse during training using the four proposed NC metrics by Zhu et al. (2024). $\mathcal{NC}_1$ measures the ratio between the within-class covariance $\mathbf{\Sigma_W}$ and the between-class covariance $\mathbf{\Sigma_B}$, $\mathcal{NC}_2$ quantifies the alignment of the learned classifier with a Simplex ETF structure, $\mathcal{NC}_3$ evaluates the duality between the classifiers $\mathbf{W}$ and the centered class-means $\mathbf{H}$, while $\mathcal{NC}_4$ measures how the bias term $\mathbf{b}$ compensates for a nonzero global mean of the features $h_G$, reflecting the degree of collapse in the bias term (refer to A.9 for detailed computations). Details on the calculation of these metrics are provided in Appendix. In Figure 4, we report the measurements of these NC metrics during standard training of a ResNet18 architecture. Larger $\gamma$

values consistently lead to higher degree of neural collapse, showing that uniform OCCE effectively amplifies this phenomenon. The same behavior is noticed for vision transformers (see A.6.2).

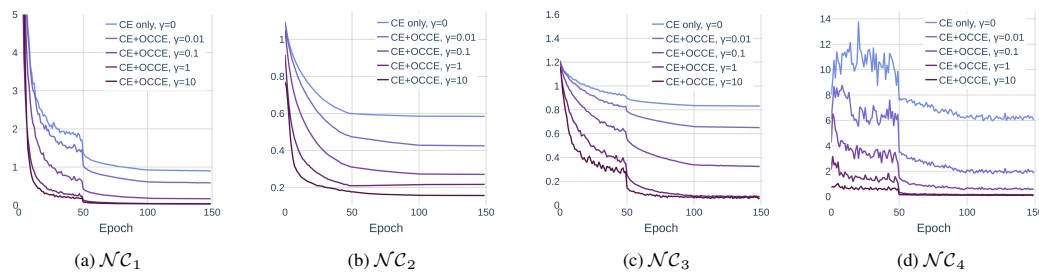

(a) $\mathcal{NC}_1$   (b) $\mathcal{NC}_2$   (c) $\mathcal{NC}_3$   (d) $\mathcal{NC}_4$

Figure 4: Uniform OCCE impact on neural collapse during training of ResNet18v2 on CIFAR-100.

**Indepedence Deficit.**   Uniform OCCE encourages each class to be equidistant from all the other classes, enforcing decoupled classes with linearly independent representations. In order to show the effect on independence deficit we follow the procedure from Feng et al. (2024) to estimate the intrinsic dimension of the penultimate classification layer required to reconstruct the original classification accuracy. We compute the covariance matrix of the representations in that layer, extract its principal components, and project onto subspaces spanned by the top-K eigenvectors.

Figure 5a shows that with standard cross entropy loss, 80%, 90%, and 99% of the original classification accuracy can be reconstructed using only 22, 37, and 64 dimensions, respectively, from the original 100-dimensional activation layer on CIFAR-100. However, when using uniform OCCE and increasing the value of $\gamma$, the reconstructed accuracy shows a more linear relationship with the dimensionality of the subspace. This suggests that uniform OCCE decouples the classes and enforces linear independence in the activation layer. For instance, with $\gamma = 10$, reconstructing 80%, 90%, and 99% of the accuracy requires 73, 89, and 98 dimensions, respectively. Intuitively, each class $c_i$ is identified by the corresponding activation $z_i$, rather than permitting predictions for $c_i$ through linear combinations of other activations $z_j, j \neq i$, directly addressing the independence deficit.

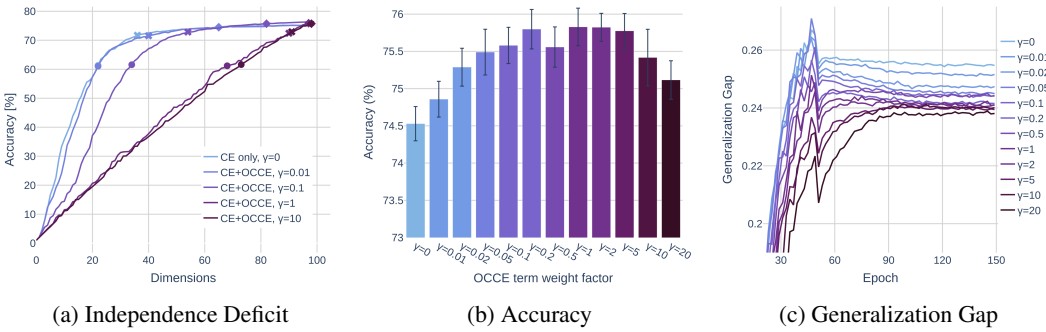

(a) Independence Deficit   (b) Accuracy   (c) Generalization Gap

Figure 5: Sensitivity analysis of the uniform OCCE objective weighting. In (a), the reconstructed accuracy is plotted based on the top-K eigenvectors of the activations layer, with $\circ$, $\times$, and $\diamond$ indicating the points where 80%, 95%, and 99% of the original accuracy is reconstructed, respectively. (b) and (c) illustrate the impact of varying $\gamma$ on validation accuracy and the generalization gap. The configuration used for these experiments is a ResNet18v2 trained on CIFAR-100, and all reported values are the average results from 10 different seeds.

**Impact on Generalization.**   In Figure 5c, we observe that increasing the value of $\gamma$ consistently reduces the generalization gap between train and test accuracy. However, as $\gamma$ becomes large, it also constrains the achievable training accuracy, as the total objective becomes more challenging to optimize, due to the increased contribution of uniform OCCE. Consequently, the network tends to converge to solutions that trade off accuracy for greater alignment of complementary classes. This suggests that there is an optimal range for $\gamma$, which depends on the dataset's difficulty and the model's complexity. For instance, in Figure 5b, a value of $\gamma = 1$ achieves the highest accuracy for ResNet18v2 on CIFAR-100. Even though this may not be the optimal value for all tasks, to maintain consistency across our following experiments, we use a value of $\gamma = 1$, unless otherwise noted.

## 3.2 CLOSED SET CLASSIFICATION EXPERIMENTS

In this section, we present the details and results of our closed-set image classification experiments, including base training and transfer learning from pretrained models.

**Base Training.** We train the following architectures: a ResNet18v2 (He et al., 2016), a MobileNetv2 (Sandler et al., 2018) and a DenseNet121 (Huang et al., 2016) on CIFAR-100 (Krizhevsky, 2009) and TinyImageNet (Le & Yang, 2015) datasets, containing 100 and 200 classes respectively. We follow a standard hyperparameter configuration across all experiments from He et al. (2015a), which involves employing an SGD optimizer with momentum of 0.9, mini-bacthes of 128 samples, weight decay of 0.0001, no dropout and a total of 150 epochs for CIFAR-100 and 90 epochs for TinyImageNet. The learning rate starts from the base value of 0.1 and is divided by 0.1 at 50 and 100 epochs for CIFAR-100, and at 30, 60 and 80 epochs for TinyImageNet. For augmentation, we follow a universal approach of normalization, random horizontal flipping and random cropping in the train set, while only normalizing and center cropping the images in the test set.

We consider different baseline losses and add the weighted uniform OCCE objective on top of them. Specifically we consider: Cross entropy loss, reverse cross entropy (Pang et al., 2017), label smoothing with $\epsilon = 0.1$ (Szegedy et al., 2015), CE combined with Negative Log loss (Kim et al., 2019), complement objective training methods (Chen et al., 2019; Kim et al., 2021), assymetric loss (Baruch et al., 2020) and a series of focal losses (Lin et al., 2017; Smith, 2022; Ghosh et al., 2024). The differentiation of our proposed approach with these competitors is presented in Section 4.

**Results.** Table 1 presents the results of base training experiments, showing pairwise comparisons between the baseline losses without and with the uniform OCCE objective. While RCE is not competitive in any of the settings, it is evident that the proposed method leads to consistent improvements along almost all baselines and models combinations, suggesting that uniform OCCE offers additional gains on top of other competitors, when used as a complementary loss. This is also visible in Figure 6, where adding OCCE improves the validation accuracy, especially after the first learning rate decay.

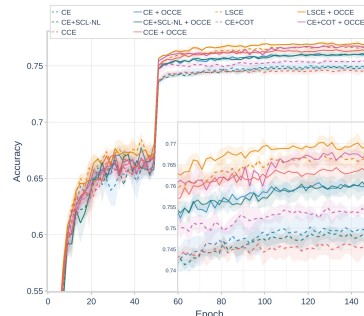

Figure 6: CIFAR-100 validation curves. Dashed lines correspond to baseline accuracy and solid lines to proposed.

Table 1: Test errors from base training using different loss functions. For each loss function, results are presented as "without OCCE ($\gamma = 0$)" (left) and "with uniform OCCE ($\gamma = 1$)" (right) separated by a slash.

| Dataset | Loss | ResNet18v2 | MobileNetv2 | DenseNet121 |
|---|---|---|---|---|
| CIFAR-100 | RCE (Pang et al., 2017) | $25.19_{\pm0.23}$ | $50.93_{\pm0.75}$ | $29.46_{\pm0.12}$ |
| | CE (Baseline) | $24.98_{\pm0.20}$ / $\mathbf{23.92}_{\pm0.23}$ | $30.03_{\pm0.35}$ / $\mathbf{28.83}_{\pm0.25}$ | $25.06_{\pm0.20}$ / $\mathbf{24.19}_{\pm0.12}$ |
| | LSCE (Szegedy et al., 2015) | $23.26_{\pm0.14}$ / $\mathbf{22.97}_{\pm0.16}$ | $29.26_{\pm0.16}$ / $\mathbf{29.14}_{\pm0.30}$ | $23.59_{\pm0.21}$ / $\mathbf{23.43}_{\pm0.06}$ |
| | CE+NL (Kim et al., 2019) | $25.10_{\pm0.14}$ / $\mathbf{23.90}_{\pm0.33}$ | $29.95_{\pm0.22}$ / $\mathbf{28.89}_{\pm0.37}$ | $25.48_{\pm0.81}$ / $\mathbf{24.71}_{\pm0.44}$ |
| | COT (Chen et al., 2019) | $24.52_{\pm0.19}$ / $\mathbf{23.21}_{\pm0.26}$ | $29.57_{\pm0.11}$ / $\mathbf{28.52}_{\pm0.32}$ | $23.70_{\pm0.75}$ / $\mathbf{22.27}_{\pm0.26}$ |
| | CCE (Kim et al., 2021) | $25.36_{\pm0.29}$ / $\mathbf{23.57}_{\pm0.22}$ | $29.81_{\pm0.25}$ / $\mathbf{28.72}_{\pm0.31}$ | $25.29_{\pm0.34}$ / $\mathbf{23.80}_{\pm0.38}$ |
| | FL (Lin et al., 2017) | $25.60_{\pm0.14}$ / $\mathbf{23.60}_{\pm0.45}$ | $30.98_{\pm0.20}$ / $\mathbf{29.45}_{\pm0.23}$ | $26.13_{\pm0.05}$ / $\mathbf{24.16}_{\pm0.19}$ |
| | ASL (Baruch et al., 2020) | $24.81_{\pm0.11}$ / $\mathbf{23.75}_{\pm0.18}$ | $29.76_{\pm0.06}$ / $\mathbf{29.10}_{\pm0.33}$ | $24.13_{\pm0.09}$ / $\mathbf{23.80}_{\pm0.20}$ |
| | CFL (Smith, 2022) | $23.69_{\pm0.27}$ / $\mathbf{23.57}_{\pm0.20}$ | $29.55_{\pm0.38}$ / $\mathbf{29.29}_{\pm0.20}$ | $24.33_{\pm0.42}$ / $\mathbf{23.81}_{\pm0.15}$ |
| | ADAFL (Ghosh et al., 2024) | $24.21_{\pm0.23}$ / $\mathbf{23.55}_{\pm0.11}$ | $34.03_{\pm0.50}$ / $\mathbf{32.86}_{\pm0.36}$ | $22.78_{\pm0.27}$ / $\mathbf{22.48}_{\pm0.10}$ |
| TinyImageNet | RCE (Pang et al., 2017) | $45.22_{\pm0.24}$ | $73.34_{\pm0.15}$ | $64.18_{\pm0.48}$ |
| | CE (Baseline) | $36.80_{\pm0.20}$ / $\mathbf{35.61}_{\pm0.13}$ | $39.69_{\pm0.49}$ / $\mathbf{38.79}_{\pm0.22}$ | $38.71_{\pm0.63}$ / $\mathbf{36.49}_{\pm0.52}$ |
| | LSCE (Szegedy et al., 2015) | $36.98_{\pm0.13}$ / $\mathbf{36.01}_{\pm0.18}$ | $39.10_{\pm0.07}$ / $\mathbf{39.08}_{\pm0.16}$ | $38.06_{\pm0.50}$ / $\mathbf{37.52}_{\pm0.66}$ |
| | CE+NL (Kim et al., 2019) | $36.40_{\pm0.10}$ / $\mathbf{35.28}_{\pm0.12}$ | $39.16_{\pm0.10}$ / $\mathbf{38.41}_{\pm0.28}$ | $38.75_{\pm1.21}$ / $\mathbf{38.66}_{\pm1.09}$ |
| | COT (Chen et al., 2019) | $35.40_{\pm0.15}$ / $\mathbf{34.74}_{\pm0.12}$ | $40.14_{\pm0.26}$ / $\mathbf{39.21}_{\pm0.26}$ | $37.60_{\pm0.21}$ / $\mathbf{35.11}_{\pm0.17}$ |
| | CCE (Kim et al., 2021) | $36.88_{\pm0.02}$ / $\mathbf{35.45}_{\pm0.04}$ | $39.19_{\pm0.35}$ / $\mathbf{38.81}_{\pm0.04}$ | $39.22_{\pm0.17}$ / $\mathbf{37.47}_{\pm0.15}$ |
| | FL (Lin et al., 2017) | $37.52_{\pm0.62}$ / $\mathbf{35.93}_{\pm0.19}$ | $40.35_{\pm0.29}$ / $\mathbf{39.41}_{\pm0.10}$ | $39.74_{\pm1.06}$ / $\mathbf{38.40}_{\pm0.68}$ |
| | ASL (Baruch et al., 2020) | $36.80_{\pm0.04}$ / $\mathbf{35.03}_{\pm0.12}$ | $39.27_{\pm0.35}$ / $\mathbf{38.35}_{\pm0.16}$ | $38.03_{\pm0.88}$ / $\mathbf{37.88}_{\pm0.31}$ |
| | CFL (Smith, 2022) | $36.69_{\pm0.33}$ / $\mathbf{35.14}_{\pm0.24}$ | $38.56_{\pm0.33}$ / $\mathbf{38.46}_{\pm0.25}$ | $37.89_{\pm0.39}$ / $\mathbf{36.74}_{\pm0.41}$ |
| | ADAFL (Ghosh et al., 2024) | $37.90_{\pm0.21}$ / $\mathbf{37.61}_{\pm0.17}$ | $45.58_{\pm0.32}$ / $\mathbf{45.34}_{\pm0.25}$ | $\mathbf{36.63}_{\pm0.19}$ / $37.16_{\pm0.23}$ |

Table 2: Test errors from transfer learning. Results are presented as "CE only ($\gamma = 0$)" (left) and "CE w/ uniform OCCE ($\gamma = 1$)" (right) separated by a slash.

| Model | Dataset | Test Errors ($\downarrow$) |
|---|---|---|
| | STL-10 | $1.87_{\pm 0.07}$ / $\mathbf{1.76}_{\pm 0.08}$ |
| WideResNet50 | CIFAR-100 | $13.58_{\pm 0.19}$ / $\mathbf{13.35}_{\pm 0.08}$ |
| | TinyImageNet | $17.33_{\pm 0.14}$ / $\mathbf{16.92}_{\pm 0.24}$ |
| | ImageNet | $21.90_{\pm 0.22}$ / $\mathbf{21.15}_{\pm 0.12}$ |
| | STL-10 | $2.27_{\pm 0.05}$ / $\mathbf{2.07}_{\pm 0.07}$ |
| Swin-T | CIFAR-100 | $12.69_{\pm 0.07}$ / $\mathbf{12.41}_{\pm 0.12}$ |
| | TinyImageNet | $14.45_{\pm 0.05}$ / $\mathbf{14.31}_{\pm 0.02}$ |
| | ImageNet | $19.69_{\pm 0.02}$ / $\mathbf{19.33}_{\pm 0.04}$ |

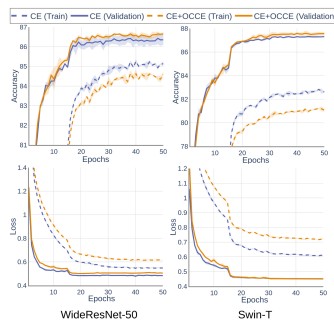

Figure 7: CIFAR-100 transfer learning validation accuracy and loss curves.

We finetune a WideResNet50 (Zagoruyko & Komodakis, 2016) and a Swin-T vision Transformer (Liu et al., 2021) on STL-10 (Coates et al., 2011), CIFAR-100, TinyImageNet and ImageNet-1k (Deng et al., 2009). We use the pretrained weights for these models provided by PyTorch, remove the classification head and initialize a new one. For hyperparameters, we use the configuration suggested by Kolesnikov et al. (2019), which involves performing 10,000 updates all medium-sized datasets, except for ImageNet where 20,000 updates are performed. SGD optimizer is employed with 0.9 momentum and a base learning rate of 0.01 and learning rate decay with a factor of 10 at 30%, 60% and 90% of the training steps. We use no weight decay and no dropout.

**Results.** Table 2 presents the results of transfer learning, while in Figure 7, we present the average performance curves for CIFAR-100 during finetuning. The loss curves reflect only the cross entropy loss in both cases. These results indicate that even though the uniform OCCE objective makes the learning task more difficult (supported by the worse performance on the training set), it results in improved generalization of the model and improved validation performance. Similar findings are reported for finetuning LLM transformers (see A.6.1).

## 3.3 OPEN SET RECOGNITION (OSR) EXPERIMENTS

In Open Set Recognition (OSR), we train the network on $K$ known classes and test with samples from both known and $U$ unknown classes. The most straightforward way to enable the reject option for the unknown classes, is through Maximum over Softmax Probabilities (MSP) (Yang et al., 2024a), where the goal is to assign lower confidence to unknown instances and higher to known ones, enabling unknown rejection by threshold.

**Decoupled Architecture.** To gain deeper insights into OCCE's behavior, we propose using decoupled projection heads for CE and OCCE. This approach allows for diversified predictions, as each head has its own projection weights, while the shared backbone is optimized by gradients from both losses (see Figure 8). This setup enables a direct comparison of linear probing under each loss within a shared representation space. We compare the results from the CE head, OCCE head, and a combination of both $\hat{\mathbf{y}}_{comb}$, by subtracting the normalized anti-class prediction $\hat{\bar{y}}_i / \max_j \hat{\bar{y}}_j$ from each class prediction $\hat{y}_i$.

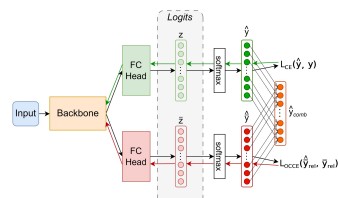

Figure 8: Decoupled architecture.

**Configuration.** We train a ResNet18v2 on FashionMNIST, CIFAR-100, and TinyImageNet for our OSR experiments. For each dataset, we split the classes into a set of known and unknown classes, denoted as $K/U$. We run each experiment 5 times with random selection of the $K/U$ classes.

**Evaluation Metrics.** Since the rarity of unknown samples is not known, OSR evaluation approaches requiring arbitrary thresholds are impractical. We use the following OSR metrics: 1. Error Rate (ER): closed-set test errors 2. Open Set Classification Rate (OSCR): trade-off between correctly classified knowns and misclassified unknowns across MSP thresholds (Chen et al., 2022). 3. Area Under the ROC curve (AUROC): the probability that knowns have higher MSP scores than unknowns.

Table 3: Average error rates, open set classification rates, and AUROC across 5 runs of our OSR configurations on FashionMNIST, CIFAR-100, and TinyImageNet datasets.

| Method | FashionMNIST (4/6) | | | CIFAR-100 (20/80) | | | TinyImageNet (40/160) | | |
|---|---|---|---|---|---|---|---|---|---|
| | ER (↓) | OSCR (↑) | AUROC (↑) | ER (↓) | OSCR (↑) | AUROC (↑) | ER (↓) | OSCR (↑) | AUROC (↑) |
| BCE | 9.66 | 68.67 | 75.93 | 13.66 | 71.19 | 80.01 | 28.99 | 59.59 | 73.63 |
| CE | 8.04 | 71.56 | 77.39 | 14.41 | 70.09 | 77.89 | 28.41 | 59.37 | 72.78 |
| RCE | 9.85 | 70.62 | 75.04 | 23.52 | 64.47 | 75.80 | 55.18 | 37.50 | 64.80 |
| CE+OCCE | 8.04 | 73.93 | 78.62 | **13.26** | 72.46 | 80.12 | **26.83** | **61.94** | **75.31** |
| Dual (CE head) | 8.26 | 72.77 | 78.48 | 13.51 | 71.09 | 78.96 | 27.58 | 60.35 | 73.72 |
| Dual (OCCE head) | **7.11** | **75.33** | **78.69** | 13.61 | **74.02** | **80.52** | 27.88 | 60.78 | 74.59 |
| Dual (Combined) | 8.22 | 74.11 | 78.51 | 13.58 | 73.75 | 80.04 | 27.38 | 61.50 | 74.51 |

**Results.** Table 3 presents the results of our OSR experiments. Employing a decoupled heads architecture enables uniform OCCE head to remain competitive across all datasets, consistently outperforming single-head CE, especially in the open-set OSCR and AUROC metrics. The CE head also benefits from the dual architecture, which indicates an improvement in the learned representations in the feature space. This is further illustrated in Figure 9, where the dual head architecture leads to tighter known class clusters and better separation from unknown classes.

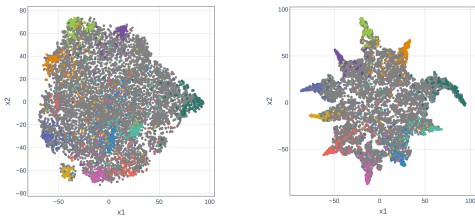

(a) CE    (b) Proposed

Figure 9: t-SNE visualization of features for CIFAR-100 validation set (K=10, U=90). Known classes are colored, while unknowns are in grey.

## 3.4 OUT-OF-DISTRIBUTION (OOD) EXPERIMENTS

We conduct additional experiments on out-of-distribution (OOD) detection, to examine if the shown improvements are agnostic to the choice of the unknown rejection algorithm. The goal here is similar to OSR, with the distinction that instead of differentiating between known and unknown classes from the same dataset, we treat all classes from an in-distribution (ID) dataset as known, while other datasets represent out-of-distribution samples.

**Configuration.** For the in-distribution datasets, we use CIFAR-100 and ImageNet-200 (full 224×224 image resolution). For out-of-distribution datasets, we employ CIFAR-10, TinyImageNet and SVHN for CIFAR-100, while SSB Hard, NINCO, and iNaturalist for ImageNet-200. All experiments are conducted using a standard ResNet-18 model. We select 10 competitive post-hoc OOD detection methods from the literature and utilized their implementations from the OpenOOD library (Yang et al., 2024a). We compare the performance of training with CE versus CE plus uniform OCCE with ($\gamma = 0.1$).

**Results.** As shown in Table 4, adding the uniform OCCE to the objective improves AUROC across all ID/OOD dataset combinations. Furthermore, CE+OCCE leads to higher accuracy on the ID datasets, achieving 77.48% for CIFAR-100 and 85.05% for ImageNet-200, compared to 76.83% and 84.28% with CE. Figure 10 illustrates that CE+OCCE outperforms CE in 9 out of 10 OOD methods on CIFAR-100 and in 8 out of 10 methods on ImageNet-200, demonstrating consistent improvements, largely agnostic to the specific post-hoc OOD method used. The full results are provided in A.7.

| ID Dataset | OOD Dataset | AUROC (↑) |
|---|---|---|
| CIFAR100 | CIFAR10 | 77.49 / **78.23** |
| | TinyImageNet | 81.25 / **82.34** |
| | SVHN | 81.52 / **83.30** |
| ImageNet200 | SSB Hard | 77.11 / **78.01** |
| | NINCO | 82.97 / **83.56** |
| | iNaturalist | 92.27 / **92.51** |

Table 4: Average AUROC across 10 post-hoc OOD methods and 3 runs. Comparisons of CE w/o and w/ OCCE ($\gamma = 0.1$) are reported, seperated with a slash.

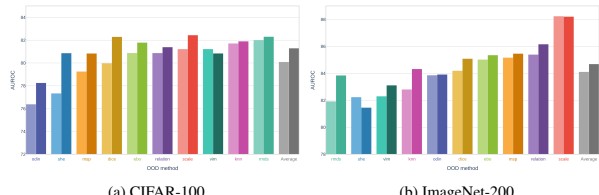

(a) CIFAR-100    (b) ImageNet-200

Figure 10: Pairwise comparisons between CE w/o OCCE (left sub-bars) and w/ OCCE (right sub-bars) across different post-hoc OOD methods. Bars represent averages over all OOD datasets and 3 runs.

## 4 RELATED WORK

**Complement Entropy.** Complement Objective Training (COT) (Chen et al., 2019) alternatively optimizes the primary cross entropy loss and a complement objective, to maximize the Shannon's entropy of complementary classes, including two updates per batch, which increases training time by 1.6-fold. Later, Kim et al. (2021) unified these objectives into Complementary Cross Entropy (CCE) loss. OCCE, unlike COT and CCE which are narrowed down to Shannon's entropy in the complementary classes, optimizes a specified anti-class distribution. As a result, OCCE can be naturally integrated with soft labels and knowledge distillation (see A.1.1).

**RCE and Label Smoothing.** Szegedy et al. (2015) introduced label smoothing to mitigate the overconfidence of neural networks, by assigning small targets to the complementary classes. However, this introduces global bias to the correct class. Pang et al. (2017) introduced RCE training procedure as a solution to this problem. RCE is an extreme case of label smoothing, when $\lambda \to \inf$, and can be seen as the stand-alone uniform OCCE objective in our framework, while our proposed approach combines class and anti-class learning, outperforming both RCE and label smoothing (see Table 1).

**ETF Simplex Classifiers.** Inspired from neural collapse, in Zhu et al. (2024), the authors fix the last layer classifier to be a Simplex ETF, demonstrating time and memory gains without performance loss. Furthermore, Simplex ETF structure has been used to mitigate minority collapse in imbalanced training (Yang et al., 2024b) and few-shot incremental learning (Yang et al., 2023). OCCE diversifies from these approaches, because it induces the Simplex ETF structure in an end-to-end learnable manner, rather than predefining the structure of the classifier.

**Hierarchical Classification.** Hierarchical classification methods have been proposed to structure class activations by leveraging parent-child relationships in predefined hierarchies (Wu et al., 2020; Valmadre, 2022; Liang & Davis, 2023). While these approaches address hierarchical relationships (vertical in a hierarchy tree), our method focuses on the flat structure, emphasizing relationships between classes at the same hierarchy level. The proposed method can be employed at multiple hierarchical levels in a similar fashion.

**Focal Approaches.** Focal Loss (Lin et al., 2017) scales the loss of well-classified samples with $(1 - \hat{y}_c)^\gamma$, where $\hat{y}_c$ is the predicted probability for the correct class, and $\gamma > 0$ emphasizes hard samples. Variants like Cyclic Focal Loss (Smith, 2022) and AdaFocal (Ghosh et al., 2024) dynamically adjust $\gamma$. These methods focus on instance-level difficulty, but they assign zero probabilities to complementary classes, effectively ignoring their structure. Introducing target anti-class distributions operates orthogonally to focal approaches and can be readily combined with them.

**Supervised Contrastive Learning.** Supervised Contrastive Learning (SupCon) (Khosla et al., 2020) extends cross entropy to the feature space by pulling positive pairs closer and pushing negative pairs farther apart. Even though it normalizes distances among both positive and negative pairs in the softmax denominator to prevent collapse, SupCon does not explicitly optimize the relationships among the complementary classes, treating them only as a normalization mechanism. This represents a fundamental difference from our proposed approach.

**Reciprocal Points Learning.** Reciprocal Points Learning (RPL) models "otherness" by positioning reciprocal points in latent space (Chen et al., 2020). Adversarial RPL (ARPL) reduces open space risk using an adversarial margin constraint (Chen et al., 2022), while ARPL + CS adds adversarial samples equidistant from all reciprocal points to simulate unseen classes. OCCE objective aligns the activations of the complementary classes for each instance, rather than relying on a single anti-class prototype for each class in the latent space.

## 5 CONCLUSIONS

In this paper, we highlight the importance of controlling the relationships between complementary classes. To this end, we introduce the concept of anti-classes as a complementary approach to modeling classification problems. By representing anti-classes using one-cold encoding, we optimize neural networks to directly learn target anti-class probability distributions through one-cold cross entropy loss. While we mainly focus on a uniform one-cold encoding, inspired by the phenomenon of neural collapse, our proposed framework can be naturally extended to non-uniform target distributions to address limitations, for example to account for inter-class relationships (see A.1.1).

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

## A APPENDIX

### A.1 LIMITATIONS OF UNIFORM OCCE

In our main experimental evaluation, we demonstrate that adding the uniform OCCE objective consistently improves performance across various settings and learning scenarios. However, this raises a natural question: should the network treat all non-target classes equally, ignoring their inherent similarities while enforcing uniformity and the Simplex ETF structure? To address this question, we explore assigning soft target anti-class distributions, that consider class similarities. Our results indicate further improvements on top of uniform OCCE, proving that our proposed framework can also efficiently support non-uniform distributions.

#### A.1.1 EXTENDING TO NON-UNIFORM ANTI-CLASS DISTRIBUTIONS

We experiment with two types of self distillation scenarios, to provide softer anti-class distributions: 1. Instance-based and 2. Class-based self-distillation.

**Instance-based self-distillation.** The new target anti-class probability distribution $\bar{\mathbf{y}}'$ for each instance is defined as a weighted combination of the uniform one-cold encoding and the network's previous predictions, as:

$$\bar{y}_i^{'} = (1 - \alpha) \cdot \bar{y}_i + \alpha \cdot \hat{\bar{y}}_i, \tag{9}$$

where $\alpha \in [0, 1]$ controls the contribution of the self-distillation. When $\alpha = 0$, the target reduces to the uniform one-cold encoding, while $\alpha = 1$ fully relies on the network's previous predictions.

**Class-based self-distillation.** In this variation, the target anti-class probability distribution $\bar{\mathbf{y}}$ incorporates class-specific self-distillation. During each epoch, we calculate the average predicted anti-class probabilities $\hat{\bar{\mathbf{y}}}$ across the entire dataset for each class. This results in a class-specific weighted matrix, $\mathbf{C}$, where we set the diagonal elements $C_{i,i} = 0$ and normalize row-wise. Here $C_{i,j}$ represents the normalized similarity of class $i$ to each complementary class $j$. The new target distribution for a sample belonging to class $i$ is a weighted combination of the uniform one-cold encoding and the class-specific weights:

$$\bar{y}_j^{'} = (1 - \alpha) \cdot \bar{y}_j + \alpha \cdot C_{i,j}, \tag{10}$$

where $\alpha \in [0, 1]$ controls the contribution of the class-specific self-distillation. When $\alpha = 0$, the target reverts to the uniform one-cold encoding, while $\alpha = 1$ incorporates only the class-specific weights.

We train ResNet18v2 on MNIST, CIFAR-10, CIFAR-100, and TinyImageNet to compare the performance of cross-entropy (CE), reverse cross-entropy (RCE), our proposed approach with uniform OCCE, and our proposed approaches using soft anti-class distributions (in Table 5). Specifically, we employ instance-based and class-based knowledge distillation techniques in the latter. The results demonstrate that while uniform OCCE consistently improves performance across all datasets, the incorporation of soft anti-class distributions provides additional improvements on top of uniform OCCE. This highlights the flexibility of OCCE, which extends beyond the uniform case to address for specific optimization objectives.

Table 5: Classification test errors of ResNet18v2. Configurations that improve upon the baseline are underlined, and the best performance for each dataset is highlighted in bold.

| Loss | MNIST | CIFAR-10 | CIFAR-100 | TinyImageNet |
|---|---|---|---|---|
| CE (Baseline) | $0.40_{\pm 0.04}$ | $5.45_{\pm 0.18}$ | $24.98_{\pm 0.40}$ | $36.80_{\pm 0.20}$ |
| RCE (Pang et al., 2017) | $0.40_{\pm 0.05}$ | $5.55_{\pm 0.16}$ | $25.19_{\pm 0.23}$ | $45.22_{\pm 0.24}$ |
| Proposed (Uniform OCCE) | $\underline{0.38}_{\pm 0.01}$ | $\underline{5.40}_{\pm 0.09}$ | $\underline{23.92}_{\pm 0.23}$ | $\underline{35.61}_{\pm 0.13}$ |
| Proposed (Instance-based self-KD, a=0.1) | $\underline{0.36}_{\pm 0.02}$ | $\underline{5.38}_{\pm 0.08}$ | $\mathbf{23.60}_{\pm 0.32}$ | $\underline{35.40}_{\pm 0.10}$ |
| Proposed (Class-based self-KD, a=0.5) | $\underline{\mathbf{0.33}}_{\pm 0.03}$ | $\underline{\mathbf{5.28}}_{\pm 0.09}$ | $\underline{23.71}_{\pm 0.10}$ | $\underline{\mathbf{35.12}}_{\pm 0.16}$ |

### A.1.2 JUSTIFICATION OF UNIFORM OCCE

Although the uniform OCCE loss does not account for the inherent relationships between classes and enforces uniformity, it still delivers consistent improvements. In this section, we analyze the uniform OCCE loss from a theoretical perspective to justify its effectiveness.

**Fisher Discriminant Ratio (FDR).** Uniform OCCE minimizes the Fisher discriminant ratio, defined as FDR $= \mathrm{tr}(S_B)/\mathrm{tr}(S_W)$, where $S_B$ and $S_W$ are the between-class and within-class scatter matrices, respectively. By minimizing within-class scatter ($S_W$) and maximizing between-class scatter ($S_B$), the loss improves class separability in the feature space.

**Information-Theoretic Perspective.** Uniform OCCE encourages the class means to form a Simplex ETF, aligning with principles of optimal coding. Observations are modeled as $h = \mu_c + z$, where $z \sim \mathcal{N}(0, \sigma^2 I)$ and $\mu_c$ represents the class means. The Simplex ETF structure maximizes the minimum pairwise distance between class means, reducing misclassification under Gaussian noise and maximizing the mutual information between the class index $c$ and the signal $h$.

**Maximum Entropy Principle.** Uniform OCCE aligns with the maximum entropy principle by assigning uniform probabilities ($\frac{1}{N-1}$) to non-target classes, which enforces the least biased distribution consistent with the constraints.

## A.2 GRADIENT DERIVATION AND CONVERGENCE BEHAVIOUR

The general cross entropy loss is given by:

$$L = -\sum_{i=1}^{N} y_i \log(\hat{y}_i) \tag{11}$$

where $\mathbf{y}$ is the target distribution and $\hat{y}_i$ is the predicted probability for class $i$, defined by the softmax function:

$$\hat{y}_i = \frac{\exp(z_i)}{\sum_{k=1}^{N} \exp(z_k)} \tag{12}$$

with $z_i$ being the activation for class $i$.

The partial derivative of the loss with respect to $\hat{y}_i$ is:

$$\frac{\partial L}{\partial \hat{y}_i} = -\frac{y_i}{\hat{y}_i} \tag{13}$$

The softmax derivative has two cases:

$$\frac{\partial \hat{y}_i}{\partial z_j} = \begin{cases} \hat{y}_i(1 - \hat{y}_i) & \text{if } i = j, \\ -\hat{y}_i \hat{y}_j & \text{if } i \neq j \end{cases} \tag{14}$$

Using the chain rule and substituting we get:

$$\frac{\partial L}{\partial z_j} = \sum_{i=1}^{N} \frac{\partial L}{\partial \hat{y}_i} \cdot \frac{\partial \hat{y}_i}{\partial z_j}$$

$$= \underbrace{\sum_{i \neq j} -\frac{y_i}{\hat{y}_i} \cdot (-\hat{y}_i \hat{y}_j)}_{i \neq j} + \underbrace{\left( -\frac{y_j}{\hat{y}_j} \cdot \hat{y}_j(1 - \hat{y}_j) \right)}_{i = j}$$

$$= \sum_{i \neq j} y_i \hat{y}_j + y_j(\hat{y}_j - 1) \tag{15}$$

Substituting the target distributions for the standard one-hot encoding $\mathbf{y}$ and our proposed one-cold encoding $\bar{\mathbf{y}}$ defined for a correct class c as:

$$\mathbf{y}_i = \begin{cases} 1 & \text{if } i = c \\ 0 & \text{if } i \neq c \end{cases} \quad \text{and} \quad \bar{\mathbf{y}}_i = \begin{cases} 0 & \text{if } i = c \\ \frac{1}{N-1} & \text{if } i \neq c \end{cases}$$

we get the derivatives of the standard cross entropy $L_{CE}$ and one-cold cross entropy $L_{OCCE}$ as:

$$\frac{\partial L_{\text{CE}}}{\partial z_i} = \begin{cases} \hat{y}_i - 1 & \text{if } i = c, \\ \hat{y}_i & \text{if } i \neq c, \end{cases} \tag{16}$$

and

$$\frac{\partial L_{\text{OCCE}}}{\partial \bar{z}_i} = \begin{cases} \hat{\bar{y}}_i & \text{if } i = c, \\ \hat{\bar{y}}_i - \frac{1}{N-1} & \text{if } i \neq c. \end{cases} \tag{17}$$

For the asymptotic behavior of cross entropy, as $\nabla_{\mathbf{z}} L_{\text{CE}} \to \mathbf{0}$, under no other constraints, we obtain:

$$\hat{y}_c = \frac{\exp(z_c)}{\sum_j \exp(z_j)} \to 1 \quad \implies \quad z_c \to +\infty, \quad \text{for the correct class } c \text{ and}$$

$$\hat{y}_k = \frac{\exp(z_k)}{\sum_j \exp(z_j)} \to 0 \quad \implies \quad z_k \to -\infty, \quad \text{for any complementary class } k.$$

In the case of one-cold cross entropy, when $\nabla_{\bar{\mathbf{z}}} L_{\text{OCCE}} \to \mathbf{0}$ and under no other constraints, we obtain:

$$\hat{\bar{y}}_c = \frac{\exp(-z_c)}{\sum_{j=1}^N \exp(-z_j)} \to 0 \quad \implies \quad \exp(-z_c) \to 0$$

$$\implies \quad z_c \to +\infty, \quad \text{for the correct class } c \text{ and}$$

$$\hat{\bar{y}}_k = \frac{\exp(-z_k)}{\sum_{j=1}^N \exp(-z_j)} \to \frac{1}{N-1}$$

$$\implies \exp(-z_k) \to \frac{1}{N-1} \sum_{j=1}^N \exp(-z_j) \approx \frac{1}{N-1} \sum_{\substack{j=1 \\ j \neq c}}^N \exp(-z_j)$$

$$\implies z_k \to -\log\left(\frac{\sum_{j \neq c} \exp(-z_j)}{N-1}\right), \quad \text{for any complementary class } k.$$

This suggests that the activations for all complementary classes $k$ converge to the same value, i.e., $z_{k_1} \to z_{k_2}$ for any $k_1, k_2 \neq c$.

## A.3 EMPIRICAL EVALUATION OF ACTIVATIONS

While the previous theoretical analysis describes the asymptotic behavior of activations under ideal conditions, in practice, the activations of a neural network often behave differently due to various factors. These include the influence of regularization methods like weight decay, which penalize large weights, and the complexity of the optimization process. Here, we provide an empirical evaluation of the distribution of activations in a converged ResNet-18 network.

In Figure 11, we present the activation distributions for both the correct classes and the complementary classes across three different training losses: a cross entropy (CE) trained network (first column), a CE+OCCE with $\gamma = 1$ trained network (second column), and an OCCE-only trained network (third column). It is evident that while the cross entropy loss results in Gaussian-like distributions for the activations of the complementary classes, similar to those of the correct classes, the introduction of the OCCE objective alters this behavior. Specifically, with uniform OCCE, all activations of the complementary classes converge to the same value. This uniformity is the primary mechanism that drives neural collapse. For each instance, the activations corresponding to its complementary classes must be equal, indicating that the instance is equidistant from all their class means. Consequently, this forces the instances within each class to collapse toward their respective class means, while ensuring that the class means themselves are equidistant in the final feature manifold.

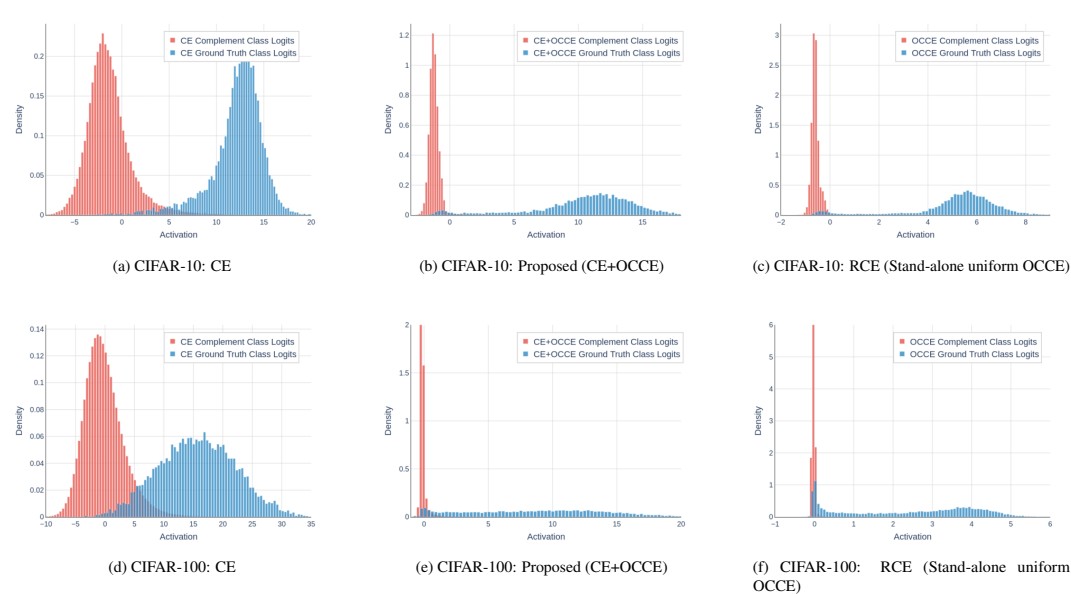

Figure 11: Activations distributions for CIFAR-10 (top row) and CIFAR-100 (bottom row), when training with CE, CE+OCCE and RCE (Stand-alone uniform OCCE). The model architecture is ResNet18v2.

## A.4 T-SNE VISUALIZATION OF FEATURE MANIFOLDS

Figure 12 presents the t-SNE visualization of the learned feature space for CIFAR-10 and CIFAR-100. Cross entropy loss (CE) appears to create Gaussian-like classes in the feature space, with some outliers scattered throughout. In contrast, RCE clusters classes in a highly specific manner. This is reflected in the t-SNE visualization, showing highly discriminative clusters for each class, where its class is forced to be equally distant from all other classes in the high-dimensional feature space. However, for the most challenging samples, RCE struggles to identify the single anti-class they do not belong to. This issue is more pronounced as the scale of the dataset increases.

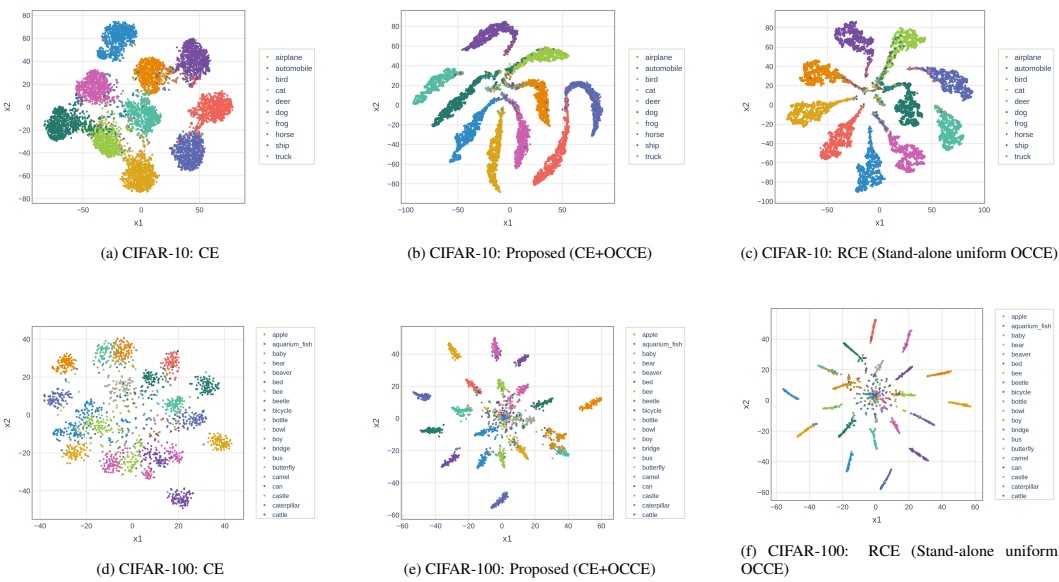

Figure 12: 2D t-SNE visualizations of the learned feature space for CIFAR-10 (first row) and CIFAR-100 (second row), when training with CE, CE+OCCE and RCE (Stand-alone uniform OCCE). For CIFAR-100 only 20 classes are shown for visualization purposes.

### A.5 CONNECTION BETWEEN OCCE AND LOW RANK REPRESENTATIONS

Empirically, we observe that uniform OCCE influences the numerical rank of each layer. Early layers exhibit higher ranks with larger $\gamma$ values, indicating more information preservation compared to CE. In contrast, the final layers have significantly lower ranks, suggesting that uniform OCCE promotes sparse, linearly independent representations for each class in the last layer.

We compute the rank of a layer function $f_i$ by calculating its Jacobian matrix $\mathbf{J_f} = (\partial f_i / \partial x_j)_{i,j} \in \mathbb{R}^{d \times n}$ for each layer $i$ over the validation set, where $d$ is the layer's output dimension and $n$ is the input dimension (Feng et al., 2024). The numerical rank is estimated by counting the singular values larger than a threshold $\epsilon = 1.19 \times 10^{-7}$. Figures 13 (left, linear scale) and (right, log scale) show the rank progression, with the linear scale revealing trends in higher ranks, and the log scale highlighting differences in lower ranks.

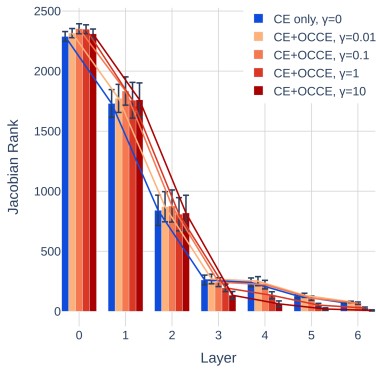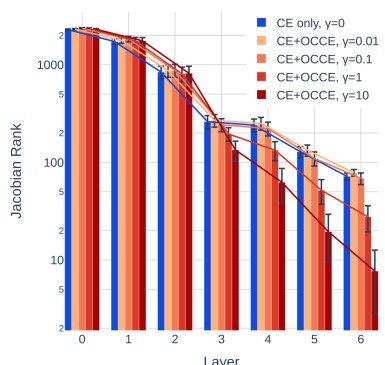

(a) Linear scale for higher ranks.      (b) Log scale for lower ranks.

Figure 13: Numerical rank approximation per layer across a ResNet18 trained on CIFAR-100, with different $\gamma$ values.

### A.6 TRANSFORMERS EXPERIMENTS

#### A.6.1 NATURAL LANGUAGE PROCESSING

To further evaluate the effectiveness of OCCE as a regularizer, we apply it to a Natural Language Processing (NLP) downstream task, specifically to Named Entity Recognition (NER). We consider the CoNLL-2003 dataset, which contains 4 entity and 1 non-entity classes (Kim et al., 2003).

For finetuning pretrained language models, we employ a standard hyperparameter configuration. This involves AdamW as optimizer, a sweep of learning rates (lr $\in 10^{-4} \cdot [1, 2, 4]$), a batch size of 32, and a total of 3 epochs of finetuning with 5 different seeds per experiment. We evaluate the overall F1-score to ensure fairness across entities. The results are shown in Table 6, where uniform OCCE consistently improves performance. Due to the small magnitude of the gains, we performed a Wilcoxon signed-rank test, which indicated the rejection of the null hypothesis with a p-value of 0.009.

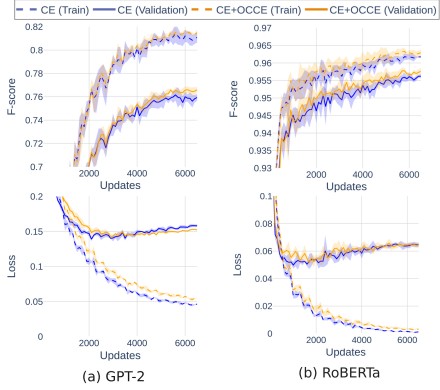

(a) GPT-2      (b) RoBERTa

Figure 14: LLMs accuracy and loss curves during finetuning on NER downstream task.

Table 6: NER F-score test errors

| Backbone | CE | CE+OCCE |
|---|---|---|
| RoBERTa | 4.38±0.05 | **4.23**±0.07 |
| BERT | 5.60±0.14 | **5.58**±0.18 |
| DistilBERT | 6.34±0.13 | **6.26**±0.20 |
| GPT-2 | 23.63±0.41 | **23.26**±0.38 |

### A.6.2 VISION TRANSFORMERS

The observed behaviors of Swin-T vision Transformer concerning neural collapse and independence deficit remain consistent with those of ResNets, supporting the architecture-agnostic nature of our findings regarding the uniform OCCE loss. Figure 15 illustrates the degree of neural collapse during training. Meanwhile, Figure 16 presents the results related to the independence deficit. The conclusions drawn are in line with our main results in Section 3.1.

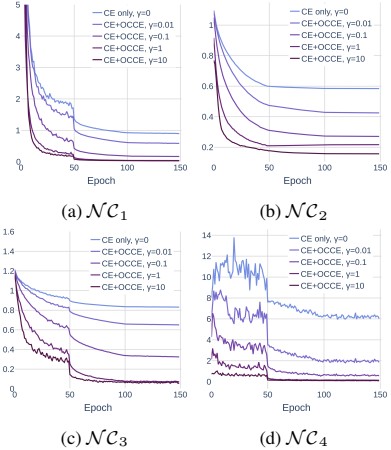

(a) $\mathcal{NC}_1$     (b) $\mathcal{NC}_2$

(c) $\mathcal{NC}_3$     (d) $\mathcal{NC}_4$

Figure 15: Neural Collapse (Swin-T).

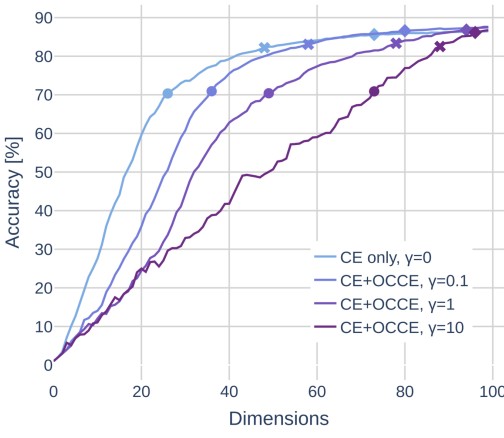

Figure 16: Independence deficit (Swin-T).

### A.7 OUT-OF-DISTRIBUTION FULL RESULTS

In this section we present the full results of our out-of-distribution experiments, where uniform OCCE leads to consistent improvements in the vast majority of ID dataset, OOD dataset and OOD method combinations. These experiments were performed using the openOOD library (Yang et al., 2024a).

Table 7: Full out-of-distribution results for ID: CIFAR-100 and OOD: CIFAR-10, TIN, SVHN. The reported values are averages of 3 runs of a ResNet18 model trained without OCCE and with uniform OCCE, separated by a slash.

| AUROC (↑) | In Distribution: CIFAR-100 | | | | |
|---|---|---|---|---|---|
| OOD Method | CIFAR-10 | TIN | SVHN | Average | $\Delta$ |
| msp | $77.89_{\pm 0.27}$ / $78.54_{\pm 0.27}$ | $81.28_{\pm 0.17}$ / $81.98_{\pm 0.05}$ | $78.58_{\pm 1.76}$ / $81.98_{\pm 0.54}$ | 79.25 / 80.83 | +1.58 |
| odin | $76.95_{\pm 0.32}$ / $77.53_{\pm 0.32}$ | $80.24_{\pm 0.17}$ / $80.51_{\pm 0.14}$ | $71.96_{\pm 3.12}$ / $76.72_{\pm 2.74}$ | 76.38 / 78.25 | +1.87 |
| rmds | $78.20_{\pm 0.40}$ / $78.52_{\pm 0.39}$ | $82.91_{\pm 0.30}$ / $82.78_{\pm 0.03}$ | $84.92_{\pm 1.11}$ / $85.61_{\pm 1.55}$ | 82.01 / 82.30 | +0.29 |
| ebo | $78.67_{\pm 0.10}$ / $79.04_{\pm 0.24}$ | $82.10_{\pm 0.28}$ / $82.72_{\pm 0.17}$ | $81.85_{\pm 2.10}$ / $83.60_{\pm 1.08}$ | 80.87 / 81.79 | +0.92 |
| vim | $75.24_{\pm 0.51}$ / $76.00_{\pm 0.42}$ | $80.58_{\pm 0.27}$ / $81.85_{\pm 0.02}$ | $87.84_{\pm 0.75}$ / $84.67_{\pm 3.79}$ | 81.22 / 80.84 | -0.38 |
| knn | $77.65_{\pm 0.21}$ / $78.03_{\pm 0.29}$ | $83.32_{\pm 0.17}$ / $83.57_{\pm 0.08}$ | $84.17_{\pm 1.45}$ / $84.10_{\pm 2.01}$ | 81.71 / 81.90 | +0.19 |
| dice | $77.20_{\pm 0.28}$ / $79.15_{\pm 0.49}$ | $79.44_{\pm 0.40}$ / $82.09_{\pm 0.18}$ | $83.28_{\pm 1.88}$ / $85.63_{\pm 1.60}$ | 79.97 / 82.29 | +2.32 |
| she | $76.90_{\pm 0.24}$ / $78.37_{\pm 0.52}$ | $77.59_{\pm 0.35}$ / $81.71_{\pm 0.26}$ | $77.52_{\pm 3.08}$ / $82.51_{\pm 1.63}$ | 77.34 / 80.86 | +3.52 |
| relation | $77.43_{\pm 0.20}$ / $77.83_{\pm 0.20}$ | $82.99_{\pm 0.25}$ / $83.35_{\pm 0.13}$ | $82.20_{\pm 2.19}$ / $82.99_{\pm 1.09}$ | 80.87 / 81.39 | +0.52 |
| scale | $78.75_{\pm 0.25}$ / $79.30_{\pm 0.29}$ | $82.01_{\pm 0.20}$ / $82.86_{\pm 0.06}$ | $82.89_{\pm 1.58}$ / $85.15_{\pm 0.71}$ | 81.22 / 82.44 | +1.22 |
| Average | 77.49 / **78.23** | 81.25 / **82.34** | 81.52 / **83.30** | 80.09 / **81.29** | +1.20 |

Table 8: Full out-of-distribution results for ID: ImageNet-200 and OOD: SSB-Hard, NINCO, iNaturalist. The reported values are averages of 3 runs of a ResNet18 model trained without OCCE and with uniform OCCE, separated by a slash.

| AUROC (↑) | In Distribution: ImageNet-200 | | | | |
|---|---|---|---|---|---|
| OOD Method | SSB-Hard | NINCO | iNaturalist | Average | $\Delta$ |
| msp | $79.29_{\pm 0.44}$ / $79.60_{\pm 0.22}$ | $84.51_{\pm 0.52}$ / $85.13_{\pm 0.33}$ | $91.70_{\pm 0.17}$ / $91.66_{\pm 0.19}$ | 85.17 / 85.46 | +0.30 |
| odin | $76.54_{\pm 0.51}$ / $76.78_{\pm 0.37}$ | $81.49_{\pm 0.13}$ / $81.84_{\pm 0.55}$ | $93.55_{\pm 0.09}$ / $93.13_{\pm 0.67}$ | 83.86 / 83.92 | +0.06 |
| rmds | $78.36_{\pm 0.24}$ / $79.29_{\pm 0.18}$ | $81.42_{\pm 0.82}$ / $83.36_{\pm 0.11}$ | $85.99_{\pm 0.77}$ / $88.89_{\pm 0.43}$ | 81.92 / 83.85 | +1.92 |
| ebo | $78.40_{\pm 0.48}$ / $79.35_{\pm 0.28}$ | $83.56_{\pm 0.19}$ / $84.07_{\pm 0.44}$ | $93.11_{\pm 0.24}$ / $92.63_{\pm 0.33}$ | 85.02 / 85.35 | +0.33 |
| vim | $75.20_{\pm 0.26}$ / $76.11_{\pm 0.34}$ | $82.61_{\pm 0.31}$ / $83.44_{\pm 0.41}$ | $89.09_{\pm 0.79}$ / $89.79_{\pm 0.90}$ | 82.30 / 83.11 | +0.81 |
| knn | $74.30_{\pm 0.47}$ / $75.30_{\pm 0.43}$ | $82.87_{\pm 0.20}$ / $84.04_{\pm 0.36}$ | $91.25_{\pm 0.67}$ / $93.63_{\pm 0.12}$ | 82.81 / 84.32 | +1.52 |
| dice | $77.02_{\pm 0.52}$ / $78.89_{\pm 0.43}$ | $81.96_{\pm 0.26}$ / $82.83_{\pm 0.33}$ | $93.60_{\pm 0.48}$ / $93.54_{\pm 0.14}$ | 84.19 / 85.09 | +0.89 |
| she | $74.31_{\pm 0.45}$ / $75.13_{\pm 0.30}$ | $79.27_{\pm 0.98}$ / $77.79_{\pm 0.37}$ | $93.13_{\pm 0.96}$ / $91.46_{\pm 0.59}$ | 82.24 / 81.46 | -0.78 |
| relation | $76.85_{\pm 0.79}$ / $78.26_{\pm 0.18}$ | $84.92_{\pm 0.78}$ / $85.98_{\pm 0.20}$ | $94.41_{\pm 0.05}$ / $94.24_{\pm 0.24}$ | 85.39 / 86.16 | +0.77 |
| scale | $80.78_{\pm 0.51}$ / $81.36_{\pm 0.24}$ | $87.05_{\pm 0.16}$ / $87.16_{\pm 0.19}$ | $96.88_{\pm 0.28}$ / $96.10_{\pm 0.11}$ | 88.24 / 88.21 | -0.03 |
| Average | 77.11 / **78.01** | 82.97 / **83.56** | 92.27 / **92.51** | 84.11 / **84.69** | +0.58 |

## A.8 Centered Kernel Alignment (CKA) Similarity Matrix

To demonstrate the diversity introduced by our proposed loss function, we use the Centered Kernel Alignment (CKA) similarity matrix to compare the representations learned by networks trained with different loss functions (Kornblith et al., 2019). CKA effectively measures the similarity between neural network representations, offering insights into how different training objectives affect the learned features.

**Centered Kernel Alignment (CKA).** CKA is computed using the formula:

$$\text{CKA}(\mathbf{K}, \mathbf{L}) = \frac{\text{HSIC}(\mathbf{K}, \mathbf{L})}{\sqrt{\text{HSIC}(\mathbf{K}, \mathbf{K}) \cdot \text{HSIC}(\mathbf{L}, \mathbf{L})}}$$

where $\text{HSIC}(\mathbf{K}, \mathbf{L})$ is the Hilbert-Schmidt Independence Criterion given by:

$$\text{HSIC}(\mathbf{K}, \mathbf{L}) = \frac{1}{(n-1)^2} \text{tr}(\mathbf{KHLH})$$

and $\mathbf{H}$ is the centering matrix defined as:

$$\mathbf{H} = \mathbf{I}_n - \frac{1}{n} \mathbf{1} \mathbf{1}^T$$

with $\mathbf{K}$ and $\mathbf{L}$ being the kernel matrices of two sets of representations, and $\mathbf{I}_n$ being the identity matrix of size $n$. Following observations from (Kornblith et al., 2019), we use a linear kernel for our experiments.

**Similarity matrix results.** Our experiments show significant divergence between the final layer representations of a network trained with cross entropy loss and one trained with our proposed loss, indicating that our loss function leads to more diverse representations. In Figure 17, we compare the similarity matrices between (a) two networks trained with cross entropy with different initialization (CE vs. CE), (b) a network trained with cross entropy versus one trained with SCL-NL loss Kim et al. (2019)(CE vs. SCL-NL), and (c) a network trained with cross entropy versus one trained with uniform OCCE (CE vs. uniform OCCE).

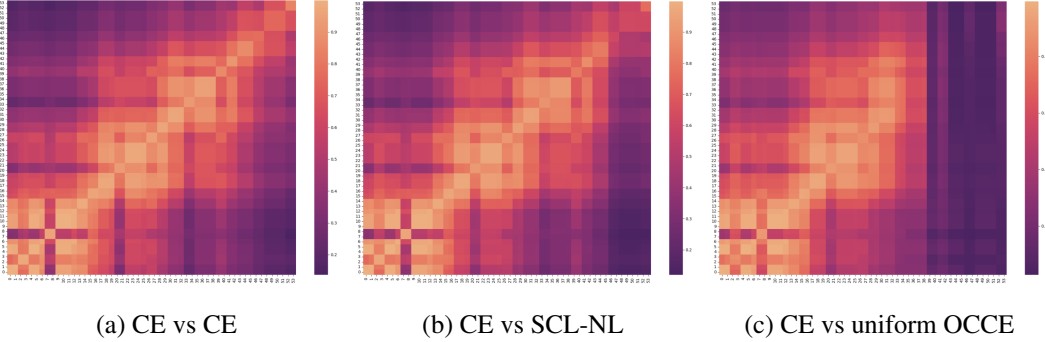

|                |                  |                       |
| :------------: | :--------------: | :-------------------: |
| (a) CE vs CE   | (b) CE vs SCL-NL | (c) CE vs uniform OCCE |

Figure 17: CKA similarity matrices between cross entropy and other losses. The architecture is ResNet18v2 and the dataset is CIFAR-100. The one-cold cross entropy loss shows the most diversified representations between the compared losses.

Two CE networks with different initialization have high similarity in their internal representations across all layers, which is showcased by the diagonal line. SCL-NL loss also have high correlation even in the final layers, because as mentioned in 4, it indirectly focuses on the correct classes. Conversely, uniform OCCE loss learns highly diversified representations towards the final layers of the networks.

## A.9 Neural Collapse Measurement Metrics

For completeness and easy reference, we detail the metrics used to evaluate the degree of neural collapse during network training. These metrics are adapted from the seminal work presented in Zhu et al. (2024).

The global mean $\mathbf{h}_G$ and class means $\mathbf{h}_k$ of the last-layer features $\{\mathbf{h}_{k,i}\}$ are defined as follows:

$$\mathbf{h}_G = \frac{1}{nK} \sum_{k=1}^{K} \sum_{i=1}^{n} \mathbf{h}_{k,i}, \quad \mathbf{h}_k = \frac{1}{n} \sum_{i=1}^{n} \mathbf{h}_{k,i} \quad \text{for } 1 \le k \le K.$$

**Within-class and Between-class Variability ($\mathcal{NC}_1$).** The within-class ($\mathbf{\Sigma}_W$) and between-class ($\mathbf{\Sigma}_B$) covariance matrices are given by:

$$\mathbf{\Sigma}_W = \frac{1}{nK} \sum_{k=1}^{K} \sum_{i=1}^{n} (\mathbf{h}_{k,i} - \mathbf{h}_k)(\mathbf{h}_{k,i} - \mathbf{h}_k)^\top, \quad \mathbf{\Sigma}_B = \frac{1}{K} \sum_{k=1}^{K} (\mathbf{h}_k - \mathbf{h}_G)(\mathbf{h}_k - \mathbf{h}_G)^\top.$$

Neural collapse within-class variability is quantified by the ratio of the trace of $\mathbf{\Sigma}_W$ to the pseudo-inverse of $\mathbf{\Sigma}_B$:

$$\mathcal{NC}_1 = \frac{1}{K} \text{trace}(\mathbf{\Sigma}_W \mathbf{\Sigma}_B^\dagger).$$

**Simplex ETF Convergence ($\mathcal{NC}_2$).** The alignment of the learned classifier $\mathbf{W}$ with a Simplex ETF is assessed by:

$$\mathcal{NC}_2 =:= \left\| \frac{\mathbf{W}\mathbf{W}^\top}{\|\mathbf{W}\mathbf{W}^\top\|_F} - \frac{1}{\sqrt{K-1}} \left( \mathbf{I}_K - \frac{1}{K} \mathbf{1}_K \mathbf{1}_K^\top \right) \right\|_F,$$

where the ETF is rescaled so that $\sqrt{\frac{1}{K-1}} \left\| \mathbf{I}_K - \frac{1}{K} \mathbf{1}_K \mathbf{1}_K^\top \right\|_F$ has unit energy (in Frobenius norm).

**Convergence to Self-duality ($\mathcal{NC}_3$).** We measure the alignment between the classifiers $\mathbf{W}$ and the centered class-means $\mathbf{H}$ to quantify the extent to which the learned features exhibit self-duality. The centered class-mean feature matrix as:

$$\mathbf{H} := [\mathbf{h}_1 - \mathbf{h}_G \ldots \mathbf{h}_K - \mathbf{h}_G] \in \mathbb{R}^{d \times K}.$$

Thus, the duality between the classifiers $\mathbf{W}$ and the centered class-means $\mathbf{H}$ is measured by:

$$NC_3 := \left\| \frac{\mathbf{W}\mathbf{H}}{\|\mathbf{W}\mathbf{H}\|_F} - \frac{1}{\sqrt{K-1}} \left( \mathbf{I}_K - \frac{1}{K} \mathbf{1}_K \mathbf{1}_K^\top \right) \right\|_F.$$

**Collapse of the Bias. ($\mathcal{NC}_4$)** In scenarios where the global mean $\mathbf{h}_G$ of the features is not zero, the bias term $\mathbf{b}$ may adjust to compensate for this global mean, effectively collapsing towards a specific direction. This phenomenon is measured by:

$$\mathcal{NC}_4 = \|\mathbf{b} + \mathbf{W}\mathbf{h}_G\|^2.$$

