# OpenReview forum: "Learning anti-classes with one-cold cross entropy loss"
_ICLR.cc/2025/Conference — Submitted to ICLR 2025_

### Official Review · Reviewer_auHK · 2024-10-28

**Soundness:** 3
**Presentation:** 3
**Contribution:** 2
**Rating:** 6
**Confidence:** 4

**Summary:**

The paper proposes to complement the standard cross-entropy loss with the idea of better controlling the behavior of predictions for negative classes, i.e., different from the target groundtruth (other classes or out-of-distribution data).

The complementary loss is expressed as the entropy of an inverse-coded problem formulation (the "one-cold" loss), where the targets are the negative classes with equal priors. The loss is claimed to favor three desirable properties: neural collapse, reduced independence deficit, and generalization.

The evaluation is carried out on problems of basic and transfer learning for classification, open set detection, and out-of-distribution detection on small to medium-sized datasets.

**Strengths:**

- The writing is clear and easy to read.

- The proposed OCCE complementary loss is simple.

- The impact of the loss on neural collapse and independence deficit is well argued and justified by experiments on some convolutional and transformer architectures.

- The positive impact on almost all experiments is consistent, although marginal for some problems (transfer learning).

**Weaknesses:**

- The impact of OCCE on generalization, while empirically effective, is less justified. It is stated that it should avoid distribution shifts (which cannot be reduced to out-of-distribution detection), but this is not really proven or demonstrated by experiments.

- The limitations of the approach are not clearly identified or summarized: the only discussion I found is about learning instabilities (l.252).

- No discussion or presentation of other known approaches for dealing with negative or hard data, e.g., contrastive or focal losses (see few references in the "Questions" section).

- Evaluation only on small or medium-sized datasets: should at least be evaluated on ImageNet.

**Questions:**

- My feeling is that favoring neural collapse and reducing the independence deficit is not always desirable for classification problems where the classes have a structure or hierarchy. Can you elaborate on this?

- Can you compare your approach with contrastive [1-4] or focal [5-7] losses?

- Some of the concurrent losses have been used for calibration [8], which is also an issue for OOD or open set recognition: what is the calibration level of the proposed approach? (can be evaluated with ECE).

[1] Khosla, P., Teterwak, P., Wang, C., Sarna, A., Tian, Y., Isola, P., ... & Krishnan, D. (2020). Supervised contrastive learning. Advances in neural information processing systems, 33, 18661-18673.

[2] Kalantidis, Y., Sariyildiz, M. B., Pion, N., Weinzaepfel, P., & Larlus, D. (2020). Hard negative mixing for contrastive learning. Advances in neural information processing systems, 33, 21798-21809.

[3] Li, S., Xia, X., Ge, S., & Liu, T. (2022). Selective-supervised contrastive learning with noisy labels. In Proceedings of the IEEE/CVF conference on computer vision and pattern recognition (pp. 316-325).

[4] J. Mukhoti, V. Kulharia, A. Sanyal, S. Golodetz, P. Torr, et P. Dokania, « Calibrating deep neural networks using focal loss », Advances in Neural Information Processing Systems, vol. 33, p. 15288‑15299, 2020.

[5] Neo, D., Winkler, S., & Chen, T. (2024). MaxEnt Loss: Constrained Maximum Entropy for Calibration under Out-of-Distribution Shift. In Proceedings of the AAAI Conference on Artificial Intelligence.

[6] X. Li et al., « Generalized Focal Loss: Learning Qualified and Distributed Bounding Boxes for Dense Object Detection », 8 juin 2020, arXiv: arXiv:2006.04388.

[7] A. Ghosh, T. Schaaf, et M. Gormley, « AdaFocal: Calibration-aware Adaptive Focal Loss », Advances in Neural Information Processing Systems, vol. 35, p. 1583‑1595, déc. 2022.

[8] C. Wang, « Calibration in Deep Learning: A Survey of the State-of-the-Art », 10 mai 2024, arXiv: arXiv:2308.01222.

---

> ### Author Response · Authors · 2024-11-20
> **Response to Reviewer auHK (1/2)**
>
> Thank you for your valuable feedback and time invested. We hope we will sufficiently address your concerns and questions and encourage a favorable reassessment of our work.
>
> **(W1)**: The impact of OCCE on generalization, while empirically effective, is less justified. It is stated that it should avoid distribution shifts (which cannot be reduced to out-of-distribution detection), but this is not really proven or demonstrated by experiments.
>
> - We are sorry for the misunderstanding caused by the reference to distribution shifts in our paper. We do not explicitly claim that OCCE improves performance under distribution shifts. The reference to distribution shifts in our paper appears in the introduction:
>
>   *"In recent work, Feng et al. (2024) also identified an independence deficit in neural networks, where the prediction confidence of some classes is redundantly determined by others through simple linear relationships, even when the classes are semantically unrelated. This entanglement can lead to overfitting and poor generalization, especially under domain shifts, as the network neglects learning distinct class representations."*
>
> - This statement draws from observations from [3], where it is stated that these phenomena are aggregated under domain shifts. However, as the reviewer correctly highlights, this is not within the scope of our paper, so we removed the reference to domain shifts, which can be confusing.
>
>
> **(W2, Q1)**: The limitations of the approach are not clearly identified or summarized: the only discussion I found is about learning instabilities (l.252). My feeling is that favoring neural collapse and reducing the independence deficit is not always desirable for classification problems where the classes have a structure or hierarchy. Can you elaborate on this?
>
> - The view that favoring neural collapse and reducing indepedence deficit is supported by recent literature [1]. However, neural collapse remains an open area of research with many unresolved theoretical and practical questions [2]. While the Simplex ETF is theoretically optimal for an abstract optimal feature design problem, this does not always guarantee better empirical performance, particularly in datasets with strong inter-class relationships or hierarchical structures.
>
> - Our work introduces a novel method to encourage and control neural collapse through the $\gamma$ factor. Our evaluations demonstrate consistent improvements across various classification tasks, including hierarchical datasets like CIFAR-100, TinyImageNet, and ImageNet. These improvements range from substantial (e.g. Table 1, Table 3) to moderate but still statistically significant, as validated by statistical tests in the cases of marginal gains (see A.5.1).
>
> - We will expand our discussion on limitations to include these open questions and highlight directions for future research.
>
> **(W3, Q2)**: No discussion or presentation of other known approaches for dealing with negative or hard data, e.g., contrastive or focal losses (see few references in the "Questions" section). Can you compare your approach with contrastive [1-4] or focal [5-7] losses?
>
> - We conducted several experiments and report the results below. An extended discussion will also be included in the related work section of the revised paper both for the contrastive learning field and focal losses.
>
> - To directly compare our method with supervised contrastive learning (SupCon), we used the official implementation from [4] and trained on CIFAR-100 with a ResNet-18. The results are as follows:
>   | Configuration             | Test Error (↓)    | GPU Time          |
>   |---------------------------|-------------------|-------------------|
>   | CE Baseline (200 epochs)  | 28.28            | 0.33 hours        |
>   | CE Baseline (500 epochs)  | 27.86            | 0.83 hours        |
>   | SupCon (1000 epochs)      | 27.17            | 11 hours          |
>   | Proposed (200 epochs)     | 27.27            | **0.33 hours**    |
>   | Proposed (500 epochs)     | **26.55**        | 0.83 hours        |
>
>     Our proposed method achieves comparable performance to SupCon with only 200 epochs, **surpassing it** when trained for 500 epochs. Importantly, our method requires only **3% and 8% of SupCon's training time** for these results.
>
> - Importantly, there is a fundamental difference between the two approaches. SupCon uses a temperature-scaled cross entropy loss, where similarities between samples of the same class appear in the numerator, and all similarities in the batch are normalized in the denominator. This aligns with the standard cross-entropy scheme, where the target class probability is normalized by the sum over all classes. In contrast, OCCE explicitly defines non-zero targets for non-target classes, thereby modeling complementary classes directly, rather than relying solely on their contribution through normalization.

---

> ### Author Response · Authors · 2024-11-20
> **Response to Reviewer auHK (2/2)**
>
> *Continuation of our answer to (W3, Q2):*
>
> - After your suggestions, we have also enriched Table 1 by including comparisons with more recent focal losses, specifically:
>   - Assymetric Loss (ASL) [6]
>   - Focal Loss: (Focal) [5]
>   - Cyclic Focal Loss: (CFL) [7]
>   - Adaptive Focal Loss: (ADAFL) [8]
>
>    For consistency, we use the official implementations of each loss and apply the same training configuration as in Table 1. We compare these losses as baselines and their performance when combined with OCCE ($\gamma$=1) on top of them. The results for CIFAR-100, averaged over 3 seeds are:
>
>     |Test Error (↓)| ResNet18v2 (Competitor / +OCCE)         | MobileNetv2 (Competitor / +OCCE)      | DenseNet121 (Competitor / +OCCE)     |
>     |-------|---------------------------------------|-------------------------------------|------------------------------------|
>     | CE    | 24.98 ± 0.20 / **23.92 ± 0.23**       | 30.03 ± 0.35 / **28.83 ± 0.25**     | 25.06 ± 0.20 / **24.19 ± 0.12**    |
>     | FL [5]   | 25.60 ± 0.14 / **23.60 ± 0.45**       | 30.98 ± 0.20 / **29.45 ± 0.23**     | 26.13 ± 0.05 / **24.16 ± 0.19**    |
>     | ASL [6]  | 24.81 ± 0.11 / **23.75 ± 0.18**       | 29.76 ± 0.06 / **29.10 ± 0.33**     | 24.13 ± 0.09 / **23.80 ± 0.20**    |
>     | CFL [7] | 23.69 ± 0.27 / **23.57 ± 0.20**       | 29.55 ± 0.38 / **29.29 ± 0.20**     | 24.33 ± 0.42 / **23.81 ± 0.15**    |
>     | ADAFL [8]| 24.21 ± 0.23 / **23.55 ± 0.11**       | 34.03 ± 0.50 / **32.86 ± 0.36**     | 22.78 ± 0.27 / **22.48 ± 0.10**    |
>
> - Experiments on TinyImageNet are ongoing and will be included in Table 1 of the revised paper.
>
> **(W4)**: Evaluation only on small or medium-sized datasets: should at least be evaluated on ImageNet.
>
> - To address this concern, we conducted additional experiments on ImageNet using a standard ResNet-50 architecture. The results are as follows:
>     | Loss      | Top-1 Test Error (↓) | Top-5 Test Error (↓) |
>     |-----------|-----------------------|-----------------------|
>     | CE        | 24.85                | 7.44                 |
>     | Proposed  | **24.24**            | **7.12**             |
>
>
> **(Q3)**: Some of the concurrent losses have been used for calibration [8], which is also an issue for OOD or open set recognition: what is the calibration level of the proposed approach? (can be evaluated with ECE).
>
> - We measured the Expected Calibration Error (ECE) of our trained ResNet-18 models on CIFAR-100 and ImageNet-200 (full resolution) as described in Section 3.4. The ECE results for CE versus our proposed method are as follows:
>   | Method     | CIFAR-100          | ImageNet-200       |
>   |------------|-----------------------|-----------------------|
>   | CE         | 0.0953 ± 0.0015      | 0.024 ± 0.0044        |
>   | Proposed   | **0.08 ± 0.0035**     | **0.018 ± 0.002**     |
>
> - While our method shows clear improvements in ECE relative to the baseline, we acknowledge that this is a byproduct of our approach rather than its primary focus.
>
>
> [1] Papyan, Vardan et al. “Prevalence of neural collapse during the terminal phase of deep learning training.” Proceedings of the National Academy of Sciences of the United States of America 117 (2020): 24652 - 24663.
>
> [2] Zhu, Zhihui et al. “A Geometric Analysis of Neural Collapse with Unconstrained Features.” Neural Information Processing Systems (2021).
>
> [3] Ruili Feng, Kecheng Zheng, Yukun Huang, Deli Zhao, Michael Jordan, and Zheng-Jun Zha. Rank diminishing in deep neural networks. In Proceedings of the 36th International Conference on Neural Information Processing Systems (NIPS '22).
>
> [4] Khosla, P., Teterwak, P., Wang, C., Sarna, A., Tian, Y., Isola, P., Maschinot, A., Liu, C., & Krishnan, D. (2020). Supervised contrastive learning. Advances in Neural Information Processing Systems
>
> [5] Lin, Tsung-Yi et al. “Focal Loss for Dense Object Detection.” IEEE Transactions on Pattern Analysis and Machine Intelligence 42 (2017): 318-327.
>
> [6] Baruch, Emanuel Ben et al. “Asymmetric Loss For Multi-Label Classification.” 2021 IEEE/CVF International Conference on Computer Vision (ICCV) (2020): 82-91.
>
> [7] Smith, Leslie. (2022). Cyclical Focal Loss. 10.48550/arXiv.2202.08978.
>
> [8] Arindam Ghosh, Thomas Schaaf, and Matt Gormley. AdaFocal: calibration-aware adaptive focal loss. In Proceedings of the 36th International Conference on Neural Information Processing Systems (NIPS '22).

---

> > ### Author Response · Authors · 2024-11-24
> > **Looking Forward to Your Thoughts**
> >
> > Dear Reviewer,
> >
> > We would like to thank you once again for your valuable feedback. We would greatly appreciate it if you could share your views on the revisions and the additional experiments we conducted to address your concerns and questions.

---

> > > ### Comment · Reviewer_auHK · 2024-11-25
> > > **Comments on authors’ rebuttal**
> > >
> > > Thank you for responding to my questions. Most of my concerns were addressed. I especially appreciate the complementary experiments on the different focal losses and supervised contrastive learning.
> > >
> > > However, from these experiments, it's hard to say that the proposed OCCE loss drastically improves performance over other losses, but at least it doesn't degrade it. I see the OCCE loss as a simple add-on that can marginally improve performance.
> > >
> > > I am inclined to revise my rating favorably.

---

> > > > ### Author Response · Authors · 2024-11-29
> > > > **Response to Reviewer auHK**
> > > >
> > > > Thank you for your response and active participation in this discussion. We would like to offer some brief clarifications based on your feedback.
> > > >
> > > > >**However, from these experiments, it's hard to say that the proposed OCCE loss drastically improves performance over other losses, but at least it doesn't degrade it.**
> > > >
> > > > To address this, we conducted statistical tests on the primary results presented in Table 1. Specifically, for each dataset, we evaluated the impact of adding our uniform OCCE objective to nine competing methods across three distinct architectures, using three fixed random seeds for each configuration. The paired t-test results are summarized as follows:
> > > > | **Dataset**         | **p-value**         | **Mean Improvement (%)** | **@90% Confidence (%)** | **@99% Confidence (%)** |
> > > > |------------------|---------------------|------------------------------------|----------------------------------|----------------------------------|
> > > > | **CIFAR-100**    | $ 3.27 \times 10^{-9} $ | 3.56                              | (2.85, 4.26)                    | (2.41, 4.71)                    |
> > > > | **TinyImageNet** | $ 3.95 \times 10^{-7} $ | 2.43                              | (1.80, 3.05)                    | (1.41, 3.44)                    |
> > > >
> > > > **The results demonstrate that the improvements are statistically significant, with p-values smaller than $10^{-6}$ for both datasets. The average relative improvements in test errors are 3.56% for CIFAR-100 and 2.43% for TinyImageNet, with tight confidence intervals, which can be considered as meaningful progress.** It is worth noting that the OCCE weight factor $\gamma$ was not optimized for each configuration to ensure simplicity and reproducibility, which could further enhance these improvements.
> > > >
> > > >
> > > > >**I see the OCCE loss as a simple add-on that can marginally improve performance.**
> > > >
> > > > Uniform OCCE loss is indeed easy and simple to use without additional computational overhead, that provides consistent improvements across diverse classification tasks, models, datasets, and losses. While the improvements may vary in magnitude, the method is robust and reliable, as evidenced in our experiments. More importantly, our proposed framework introduces a novel idea: integrating anti-class learning into classification tasks to control the relationships between complementary classes. This framework also supports non-uniform target distributions, which, depending on the dataset and task, can offer even greater improvements. Please refer to Section A.1: *Limitations of Uniform OCCE* (l.704) for a detailed discussion and additional experiments demonstrating the benefits of non-uniform distributions.
> > > >
> > > > >**I am inclined to revise my rating favorably.**
> > > >
> > > > We are excited to hear that you are considering raising your score based on our clarifications, and we hope this response sufficiently addresses any remaining concerns. If you have additional questions or would like further clarifications, we would be more than happy to provide them.

---

### Official Review · Reviewer_Q6sr · 2024-11-03

**Soundness:** 2
**Presentation:** 3
**Contribution:** 2
**Rating:** 5
**Confidence:** 4

**Summary:**

The paper presents "one-cold cross entropy loss" (OCCE), as opposed to the widely-adopted "one-hot cross entropy loss" (CE). The motivation is that "the typical CE loss primarily focuses on the ground-truth classes, ignoring the relationships between the non-target, complementary classes. This leaves valuable information unexploited during optimization." Therefore, the paper propose to encode all the non-target classes as label-1, so-called "anti-class", and the target as label-0. As stated in the paper, this OCCE equally treats all non-target classes,  out-of-distribution samples, noise, and in general any instance that does not belong to the true class. The OCCE loss equally distributes activations across all non-target classes. Experiments on CIFAR-100 and TinyImageNet demonstrate that OCCE performs better than CE and other CE variants, yields better performance on open-set recognition and out-of-distribution detection.

**Strengths:**

- It is intersting to see that the paper considers a different way to encode ground-truth information to train classification models, i.e., using the OCCE loss rather than the typical one-hot cross-entropy loss.
- The writing is good and the readability is good.
- Experiments cover multiple aspects including closed-set classification, open-set recognition, and out-of-distribution detection.

**Weaknesses:**

- As the paper considers to encode the ground-truth differently from one-hot, by using the proposed one-cold encoding, it is expected to compare other losses such as Supervised Contrastive Loss (SupCon) [R1]. SupCon has supervisions from within-class positive pairs and between-class negative pairs. It can be thought of treating non-target classes equally. It is straightforward to ask whether the OCCE loss has advantages over SupCon.

- The paper combines OCCE loss and the typical CE loss in Eq. (8) to train models, and explains that "we empirically observed convergence instabilities on more complex datasets with a larger number of classes, due to the challenge of perfectly aligning the activations of all complementary classes". It suggests that it is factually unreasonable to treat non-target classes equally, make supervisions from them while ignoring the target-class

- There are some more related works that explore hierarchical classification [R2,R3,R4], which seems to move forward beyond "treating non-target classes equally". These methods seem to satisfy the motivation of the paper "structuring the activations of these complementary classes". The paper should carry out more careful survey, discuss the importance and difference from related works.

- The datasets (CIFAR100 and TinyImageNet) and network architectures (ResNet18v2) used in the experiments are too small. Related works consider larger-scale datasets (e.g., ImageNet) and more diverse larger networks (e.g., ResNet-200) in experiments to demonstrate the effectiveness of proposed methods in nowadays' standards. The paper should enrich experiments.

Citations to compare
- [R1] Supervised contrastive learning, NeurIPS, 2020
- [R2] Large margin hierarchical classification, ICML, 2004
- [R3] Hierarchical classification: combining Bayes with SVM, ICML 2006
- [R4] Hierarchical classification at multiple operating points, NeurIPS, 2022

**Questions:**

Authors are encouraged to address the weaknesses in rebuttal/responses. Please refer to the weaknesses for details.

---

> ### Author Response · Authors · 2024-11-20
> **Response to Reviewer Q6sr (1/2)**
>
> Thank you for your valuable feedback and time invested. We hope we will sufficiently address your concerns and questions and encourage a favorable reassessment of our work.
>
> **(W1)**: As the paper considers to encode the ground-truth differently from one-hot, by using the proposed one-cold encoding, it is expected to compare other losses such as Supervised Contrastive Loss (SupCon) [R1]. SupCon has supervisions from within-class positive pairs and between-class negative pairs. It can be thought of treating non-target classes equally. It is straightforward to ask whether the OCCE loss has advantages over SupCon.
>
> - We thank you for the suggestion and we conducted experiments to directly compare the two methods. Using the official implementation from [1], we trained on CIFAR-100 with a ResNet-18, and report the results below:
>
>   | Configuration             | Test Error (↓)    | GPU Time          |
>   |---------------------------|-------------------|-------------------|
>   | CE Baseline (200 epochs)  | 28.28            | 0.33 hours        |
>   | CE Baseline (500 epochs)  | 27.86            | 0.83 hours        |
>   | SupCon (1000 epochs)      | 27.17            | 11 hours          |
>   | Proposed (200 epochs)     | 27.27            | **0.33 hours**    |
>   | Proposed (500 epochs)     | **26.55**        | 0.83 hours        |
> - In this setup, **our proposed method nearly matches the performance of SupCon in just 200 epochs and surpasses it when trained for 500 epochs, in only 3\% and 8\% of SupCon training time**, respectively. Additionally, we would like to highlight an important fundamental difference. SupCon uses a temperature-scaled cross entropy loss, where similarities between samples of the same class appear in the numerator, normalized by all similarities in the batch in the denominator. This follows the standard cross entropy scheme, where the target class sample is in the numerator and normalized by the sum over all classes in the denominator. In contrast, our approach sets explicit non-zero targets for non-target classes.
>
> - We will revise the Related Work section to provide a detailed discussion of the differences between SupCon and our proposed method.
>
> **(W2)**: The paper combines OCCE loss and the typical CE loss in Eq. (8) to train models, and explains that "we empirically observed convergence instabilities on more complex datasets with a larger number of classes, due to the challenge of perfectly aligning the activations of all complementary classes". It suggests that it is factually unreasonable to treat non-target classes equally, make supervisions from them while ignoring the target-class.
>
> - We do not propose OCCE as a replacement for cross entropy loss but rather as a complementary part of the loss function. While the presentation emphasizes the contrast between the two losses, this is to highlight their unique differences. **We will further clarify that combining the two losses is essential to achieve improvements. When the target class is included, treating non-target classes equally and incorporating supervision from them proves effective, even for datasets with 100, 200 or 1000 classes.**

---

> > ### Author Response · Authors · 2024-11-20
> > **Response to Reviewer Q6sr (2/2)**
> >
> > **(W3)**: There are some more related works that explore hierarchical classification [R2,R3,R4], which seems to move forward beyond "treating non-target classes equally". These methods seem to satisfy the motivation of the paper "structuring the activations of these complementary classes". The paper should carry out more careful survey, discuss the importance and difference from related works.
> >
> > - Hierarchical classification is indeed a relevant field, and we will include a discussion in the Related Work section to address its connection to our paper. While hierarchical approaches focus on structuring activations of different classes vertically (i.e., between parent and child nodes in a class hierarchy), our approach is primarily concerned with the flat structure, emphasizing relationships between classes at the same hierarchical level. Our work, while not specifically targeting hierarchical classification, can be extended to handle such cases. The consistent improvements in leaf classification accuracy for naturally hierarchical datasets (CIFAR-100 and ImageNet) support this potential application of our approach.
> >
> > **(W4)**: The datasets (CIFAR100 and TinyImageNet) and network architectures (ResNet18v2) used in the experiments are too small. Related works consider larger-scale datasets (e.g., ImageNet) and more diverse larger networks (e.g., ResNet-200) in experiments to demonstrate the effectiveness of proposed methods in nowadays' standards. The paper should enrich experiments.
> >
> > - To address this concern, we conducted additional experiments on ImageNet using a standard ResNet-50 architecture, which is commonly used in literature. The results are as follows:
> >     | Loss      | Top-1 Test Error (↓) | Top-5 Test Error (↓) |
> >     |-----------|-----------------------|-----------------------|
> >     | CE        | 24.85                | 7.44                 |
> >     | Proposed  | **24.24**            | **7.12**             |
> >
> > - We also run additional experiments following the suggestions of other reviewers, comparing with more recent methods and in long tail settings.
> > -  Apart from ResNet18v2 we have also conducted experiments using DenseNet121, WideResNet-50 and a Swin-T vision Transformer to provide a more diverse evaluation of our method and to show that our proposed method is agnostic to the employed architecture.
> >
> > [1] Khosla, P., Teterwak, P., Wang, C., Sarna, A., Tian, Y., Isola, P., Maschinot, A., Liu, C., & Krishnan, D.  Supervised contrastive learning. Advances in Neural Information Processing Systems (2020).

---

> > > ### Author Response · Authors · 2024-11-24
> > > **Looking Forward to Your Thoughts**
> > >
> > > Dear Reviewer,
> > >
> > > We would like to thank you once again for your valuable feedback. We would greatly appreciate it if you could share your views on the revisions and the additional experiments we conducted to address your concerns and questions.

---

> > > > ### Comment · Reviewer_Q6sr · 2024-11-27
> > > > **thanks**
> > > >
> > > > Thank you for the rebuttal and more experiments. They partially address my concerns. I appreciate the new results but I would expect to all see more analysis and discussions, e.g., explaining why the proposed loss is superior over SupCon other than experimental results. I would also like to see all the details in the experimental comparison which would help me and others understand if there are implementation details making one is better than the other. I am not convinced that it is reasonable to assume all negative classes should be treated equally; rather, I tend to believe that some classes are more similar than others to a target class (e.g., bird is more similar to airplane w.r.t shape than frog, horse is more similar to frog w.r.t having four legs than bird, etc.). By the way, hierarchical classification can treat all the classes of interest in a "flat" manner by placing them in leaves. Therefore, I maintain my rating.

---

> > > > > ### Author Response · Authors · 2024-11-29
> > > > > **Response to Reviewer Q6sr (1/2)**
> > > > >
> > > > > Thank you for your detailed feedback. We completely agree that your concerns are reasonable, and we will make every effort to address each point you have raised. We kindly invite you to review our response and the updated manuscript (we highlight the relevant revised parts in this response). We hope our response will address your concerns effectively and convince you to consider raising your score.
> > > > >
> > > > > >**I am not convinced that it is reasonable to assume all negative classes should be treated equally; rather, I tend to believe that some classes are more similar than others to a target class (e.g., bird is more similar to airplane w.r.t shape than frog, horse is more similar to frog w.r.t having four legs than bird, etc.).**
> > > > >
> > > > > We agree with you on this point. **We do not argue that the uniform OCCE is the best anti-class distribution one can define.** Its simplicity and directness was the reason behind being the main focus in this paper, as the goal was to convey the general idea of anti-classes, demonstrate consistent improvements, and showcase the intriguing connections to neural collapse and independence deficit we discovered. To further explore the direction you suggest, we conducted additional experiments with soft anti-class target distributions that account for inter-class similarities. Specifically, we experimented with two variations:
> > > > > - **Instance-based self-distillation**
> > > > > - **Class-based self-distillation**
> > > > >
> > > > > **Both approaches utilize a soft non-uniform one-cold encoding, that does not treat all negative classes equally**. The first accounts for instance-dependent similarities to each complementary class, while the second considers dataset-dependent similarities between classes. For implementation details, please refer to our revised Section A.1: *Limitations of Uniform OCCE* (l.704).
> > > > >
> > > > > | **Loss**                           | **MNIST**            | **CIFAR-10**        | **CIFAR-100**           | **TinyImageNet**        |
> > > > > |------------------------------------|----------------------|---------------------|-------------------------|-------------------------|
> > > > > | **CE (Baseline)**                  | 0.40 ± 0.04          | 5.45 ± 0.18         | 24.98 ± 0.40            | 36.80 ± 0.20            |
> > > > > | **Proposed (Uniform OCCE)**        | 0.38 ± 0.01          | 5.40 ± 0.09         | 23.92 ± 0.23            | 35.61 ± 0.13            |
> > > > > | **Proposed (Instance-based self-KD, a=0.1)** | 0.36 ± 0.02       | 5.38 ± 0.08         | **23.60 ± 0.32**        | 35.40 ± 0.10            |
> > > > > | **Proposed (Class-based self-KD, a=0.5)**    | **0.33 ± 0.03**    | **5.28 ± 0.09**     | 23.71 ± 0.10            | **35.12 ± 0.16**        |
> > > > >
> > > > > The best performance for each dataset is highlighted in bold. **It is evident that the soft OCCE methods outperform the uniform OCCE approach, which proves that our proposed approach effectively supports non-uniform target anti-class distributions. However, even the uniform OCCE results in consistent improvements over cross entropy.** To justify this behavior, we included a discussion in Section A.1.2: *Justification of Uniform OCCE* (l.756), where we attribute the improvements of uniform OCCE to the optimized Fisher discriminant ratio, optimal coding, and the maximum entropy principle.
> > > > >
> > > > > >**I appreciate the new results but I would expect to all see more analysis and discussions, e.g., explaining why the proposed loss is superior over SupCon other than experimental results.**
> > > > >
> > > > > Let us present in more depth the fundamental differences of our proposed method and SupCon [1]. First, SupCon uses the infoNCE loss, where the gradient contribution of negative pairs diminishes logarithmically as the number of classes $K$ increases (log-$K$ curse of contrastive learning). The infoNCE loss is defined as:
> > > > >
> > > > > $$
> > > > > L_{\text{SupCon}} = \sum_{i \in I} \frac{-1}{|P(i)|} \sum_{p \in P(i)} \log \frac{\exp(\text{sim}(z_i, z_p) / \tau)}{\sum_{a \in N(i)} \exp(\text{sim}(z_i, z_a) / \tau)},
> > > > > $$
> > > > >
> > > > > where $\text{sim}(z_i, z_n)$ represents the similarity function (e.g., cosine similarity), and $N(i)$ is the set of all negatives for anchor $i$. The gradient from a negative sample $j \in N(i)$ is proportional to:
> > > > >
> > > > > $$
> > > > > \frac{\exp(\text{sim}(z_i, z_j) / \tau)}{\sum_{k \neq i} \exp(\text{sim}(z_i, z_k) / \tau)}.
> > > > > $$
> > > > >
> > > > > As $K$ grows, the denominator increases, leading to diminishing gradient contributions from individual negatives. In contrast, cross entropy approaches do not suffer from this issue. The gradient for a complementary class ($j \neq i$) is:
> > > > >
> > > > > $$
> > > > > \frac{\partial L_{\text{CE}}}{\partial z_j} = \hat{y}_j.
> > > > > $$
> > > > >
> > > > > This ensures that complementary classes consistently contribute to the optimization through their probabilities, even for large $K$. However, the issue here is that when all complementary probabilities $\hat{y}_j$ are small, the gradient contributions from them also become small—not due to uniformity or convergence to a desired distribution, but because $\hat{y}_i$ (the correct class probability) approaches 1.
> > > > >
> > > > > *(continues)*

---

> > > > > > ### Author Response · Authors · 2024-11-29
> > > > > > **Response to Reviewer Q6sr (2/2)**
> > > > > >
> > > > > > *Continuation of the previous response:*
> > > > > >
> > > > > > On the other hand, the proposed OCCE loss extends CE by explicitly assigning non-zero targets \(y_{\bar{i}}^j > 0\) to complementary classes. The gradient is:
> > > > > >
> > > > > > $$
> > > > > > \frac{\partial L_{\text{OCCE}}}{\partial z_j} = \hat{\bar{y}}_j - \bar{y}_i^j
> > > > > > $$
> > > > > >
> > > > > > where $\bar{y}_{i}^j$ is the anti-class target for the complementary class $j$ (e.g. $\frac{1}{N-1}$ if we use a uniform OCCE). **These gradients provide a stronger training signal for the complementary classes in scenarios where both cross entropy and SupCon fail to do so. This leads to considerably faster convergence for OCCE compared to SupCon. This is also acknowledged by the authors of SupCon, since they use at least twice the number of epochs for convergence. Furthermore, our proposed approach enables optimization towards a desirable target distribution for the complementary classes through the anti-class formulation, whether this is chosen to be a uniform or a non-uniform distribution. This ability is not provided by SupCon.**
> > > > > >
> > > > > > >**I would also like to see all the details in the experimental comparison which would help me and others understand if there are implementation details making one is better than the other.**
> > > > > >
> > > > > > We conducted experiments using the official GitHub repository provided by the authors of SupCon:
> > > > > > https://github.com/HobbitLong/SupContrast
> > > > > >
> > > > > > This repository includes implementations of both SupCon and standard cross entropy training. Training SupCon on a pytorch resnet18 required a total of 11 hours to complete the 1000 pretraining epochs specified by the authors. We did not modify or finetune any hyperparameters and strictly followed the original setup proposed by the authors. For comparison, we trained CE and CE+uniform OCCE for 200 and 500 epochs under the same fixed random seed.
> > > > > >
> > > > > > In all main experiments, we follow best practices from the literature, which are appropriately cited in our paper, and the official implementations of our compared losses. To ensure reproducibility, we ran at least three independent trials with the same configuration and shared fixed seed for each setup. Additionally, for the sensitivity analysis (Figure 5, l.300), we expanded the runs to 10 seeds to provide an even more robust evaluation. **Our results are reproducible and the code will be publicly available.**
> > > > > >
> > > > > > >**By the way, hierarchical classification can treat all the classes of interest in a "flat" manner by placing them in leaves.**
> > > > > >
> > > > > > We agree that hierarchical classification can be configured to treat all classes in a "flat" manner by placing them as leaves, and in this case the proposed method can be trivially applied in the same way. However, most hierarchical methods that demonstrate performance improvements explicitly leverage parent-child relationships. For instance, approaches such as [2,3,4,5] utilize these hierarchical dependencies to enhance classification.
> > > > > >
> > > > > > Our proposed loss can be also adopted in such hierarchical classification scenarios. Recent works [6] have shown that inducing a Simplex ETF structure in hierarchical datasets can lead to improved performance, aligning well with the geometric principles our approach encourages. We updated the Related Work section to discuss hierarchical classification methods (l.508).
> > > > > >
> > > > > >
> > > > > > [1] Prannay Khosla, et al. 2020. Supervised contrastive learning. In Proceedings of the 34th International Conference on Neural Information Processing Systems (NIPS '20)
> > > > > >
> > > > > > [2] Wu, C.J., Tygert, M., & LeCun, Y. A hierarchical loss and its problems when classifying non-hierarchically. PLoS ONE. (2017)
> > > > > >
> > > > > > [3] Redmon, Joseph and Ali Farhadi. “YOLO9000: Better, Faster, Stronger.” 2017 IEEE Conference on Computer Vision and Pattern Recognition (CVPR) (2017)
> > > > > >
> > > > > > [4] Bertinetto, Luca et al. “Making Better Mistakes: Leveraging Class Hierarchies With Deep Networks.” IEEE/CVF Conference on Computer Vision and Pattern Recognition (CVPR) (2020)
> > > > > >
> > > > > > [5] Tz-Ying Wu, Pedro Morgado et al. Solving Long-Tailed Recognition with Deep Realistic Taxonomic Classifier. In Computer Vision – (ECCV) (2020)
> > > > > >
> > > > > > [6] Liang, Tong and Jim Davis. “Inducing Neural Collapse to a Fixed Hierarchy-Aware Frame for Reducing Mistake Severity.” IEEE/CVF International Conference on Computer Vision (ICCV) (2023)

---

> > > > > > > ### Author Response · Authors · 2024-12-01
> > > > > > > **Reminder: Rebuttal Discussion Period Closing Soon**
> > > > > > >
> > > > > > > Dear Reviewer,
> > > > > > >
> > > > > > > As the rebuttal discussion period is nearing its end, we kindly remind you of our response to your concerns. We have thoroughly addressed every aspect of your feedback and believe we have sufficiently clarified the points raised. We would greatly appreciate hearing whether our response has addressed your concerns. Thank you again for your time and consideration.

---

> ### Comment · Area_Chair_MBF4 · 2024-11-27
>
> Dear reviewer,
>
> Today is the last day for reviewers to ask questions to authors. Did the authors' rebuttal address your concern? Do you have any additional questions?

---

### Official Review · Reviewer_gKu8 · 2024-11-03

**Soundness:** 2
**Presentation:** 3
**Contribution:** 2
**Rating:** 3
**Confidence:** 4

**Summary:**

While softmax cross entropy loss is commonly used for supervised classification, it overlooks the relationships between non-target classes, leading to underutilized information. To address this, the authors introduce one-cold cross entropy (OCCE) loss, which targets the activations of complementary classes by defining an anti-class for each target class, encompassing all non-target instances. By encouraging the model to uniformly distribute activations across non-target classes, OCCE loss promotes a more structured and symmetric feature space, enhancing neural collapse and addressing the independence deficit problem. The experiments demonstrate that OCCE loss consistently improves performance across various settings.

**Strengths:**

1. This paper is easy to follow.

2. The effect of OCCE loss on the occurrence of neural collapse and Indepedence Deficit is explored, which is good.

**Weaknesses:**

1. The proposed anti-class is actually  reverse cross entropy (RCE) adopted in [1]. There is nothing new. In [1], the authors also utilized CE+\lambda RCE like Enq (8) in this paper. So the novelty of this paper is quite limited.

2. The compared baselines are quite out of date.


[1] Tianyu Pang, Chao Du, Yinpeng Dong, Jun Zhu. Towards Robust Detection of Adversarial Examples. NeurIPS 2018.

**Questions:**

Please see the Weakness.

---

> ### Author Response · Authors · 2024-11-20
> **Response to Reviewer gKu8 (1/2)**
>
> Thank you for your valuable feedback and time invested. We hope we will sufficiently address your concerns about the contribution of this work and encourage a favorable reassessment of our work.
>
> **(W1)**: The proposed anti-class is actually reverse cross entropy (RCE) adopted in [1]. There is nothing new. In [1], the authors also utilized CE+$\lambda$RCE like Enq (8) in this paper. So the novelty of this paper is quite limited.
>
> We thank the reviewer for pointing out this relevant work [1]. We appreciate the opportunity to clarify the unique contributions of our paper compared to [1], and revise our paper to discuss it. Below we outline the distinct contributions of our paper in regard with the motivation, technical contribution and experimental evaluation:
>
> - **Motivation**:
>    - [1] focuses on adversarial robustness and define the entropy of normalized non-maximal predictions (non-ME) as an adversarial attack detection metric. Then they introduce RCE training procedure as a practical solution to the bias that label smoothing introduces to the correct class.
>    - Our motivation begins from the limited role of the complementary classes in the cross entropy loss, where their participation is only due to the softmax operation. **We introduce the concept of an anti-class, to allow us to model the complementary class space and explicitly define non-zero targets for the complementary class neurons, to enhance the gradient contribution from them.**
>    - **The uniform anti-class distribution used in our experiments is only a specific implementation of this broader idea, which can not be reduced to the RCE scheme.** Furthermore, the proposed approach enables using more complex learning architectures, increasing flexibility and leading to improved performance in various scenarios (closed set, open set, long tail classification, and out-of distribution detection).
>
> - **Technical contributions**:
>   - **Equation 4 of [1], given as $L^\lambda_\text{CE}(x, y) = L_\text{CE}(x, y) - \lambda \cdot R_y^\top \log F(x)$ is the label smoothing equation introduced in [2], with smoothing factor $\lambda$, and is not the same with our Equation (8).** More specifically, as the authors of [1] mention, RCE is an extreme case of label smoothing when $\lambda \to \inf$, while our Equation (8) combines RCE and CE in a single objective. Please note that [1] proposes RCE only as a substitute of CE (not combining them both), leading to significant training difficulties as the scale of dataset increases.
>   - Let us decompose the function $F(x) = (G \circ S)(x)$, where $G$ is the mapping until the penultimate layer and $S$ is the softmax operation. Then Equation (4) from [1] can be written as: $L^\lambda_\text{CE}(x, y) = L_\text{CE}(x, y) - \lambda \cdot R_y^\top \log[S(G(x)))]$.
>   - Our proposed loss is defined as $L(x, y) = L_\text{CE}(x, y) + \gamma \cdot R_y^\top \log[S(-G(x))]$. Here, the negation of the logits before the softmax results in a completely different objective. **This has not been proposed before and is actually the only way to successfully include RCE in classification tasks in realistic datasets, as shown in the following Table (refer to A.3).** Note that when trained in ImageNet, RCE converges to a uniform distribution among all classes, which leads to substantially lower classification accuracy than standard cross entropy.
>
>       | Test Error (↓) | MNIST           | CIFAR-10       | CIFAR-100        | TinyImageNet     |
>       |----------------|-----------------|----------------|------------------|------------------|
>       | CE             | 0.40 ± 0.04     | 5.45 ± 0.18    | 24.98 ± 0.40     | 36.80 ± 0.20     |
>       | RCE            | 0.40 ± 0.05     | 5.55 ± 0.16    | 25.19 ± 0.23     | 45.22 ± 0.24     |
>       | Proposed       | **0.38 ± 0.01** | **5.40 ± 0.09**| **23.92 ± 0.23** | **35.61 ± 0.13** |
>
>   - Moreover, we extend this by decoupling the linear projections for CE and OCCE, allowing for more flexibility, resulting in further improvements in performance (see Figure 7 and Table 3).
>
> [1] Pang, Tianyu et al. “Towards Robust Detection of Adversarial Examples.” Neural Information Processing Systems (2017).
>
> [2] Szegedy, Christian et al. “Rethinking the Inception Architecture for Computer Vision.” IEEE Conference on Computer Vision and Pattern Recognition (CVPR) (2015).

---

> > ### Author Response · Authors · 2024-11-20
> > **Response to Reviewer gKu8 (2/2)**
> >
> > *Continuation of our answer to (W1):*
> >
> > - **Experiments**:
> >    - [1] focuses only on the robustness against adversarial attacks on simple datasets (MNIST and CIFAR10).
> >    - We consider different classification scenarios and show that our proposed combined loss leads to consistent improvements, **agnostic** to the employed classification loss (Table 1 and Table 3), or post-hoc out-of-distribution detection method (Table 4), (indicative results are provided in the table above). Following the suggestion of Reviewer 619x, we also demonstrated that the proposed method leads to improvements in long tail classification.
> >    - We reveal a **direct connection of the weight factor of OCCE ($\gamma$) with neural collapse, independence deficit and generalization, that has not been explored before.**
> >
> > - Based on these, we believe that [1] does not reduce the contribution of this work, rather than showing one extra application area, i.e. adversarial robustness. We revise our paper to include a discussion of [1], following these observations.
> >
> > **(W2)**: The compared baselines are quite out of date.
> >
> > - We conducted additional experiments, following suggestions from other reviewers, in order to make more up to date comparisons. More specifically, we enrich Table 1 the results by considering comparisons with [3],[4],[5] and [6]. The results for CIFAR-100 are presented below:
> >
> >     |Test Error (↓)| ResNet18v2 (Competitor / +OCCE)         | MobileNetv2 (Competitor / +OCCE)      | DenseNet121 (Competitor / +OCCE)     |
> >     |-------|---------------------------------------|-------------------------------------|------------------------------------|
> >     | CE    | 24.98 ± 0.20 / **23.92 ± 0.23**       | 30.03 ± 0.35 / **28.83 ± 0.25**     | 25.06 ± 0.20 / **24.19 ± 0.12**    |
> >     | FL [3]  | 25.60 ± 0.14 / **23.60 ± 0.45**       | 30.98 ± 0.20 / **29.45 ± 0.23**     | 26.13 ± 0.05 / **24.16 ± 0.19**    |
> >     | ASL [4]  | 24.81 ± 0.11 / **23.75 ± 0.18**       | 29.76 ± 0.06 / **29.10 ± 0.33**     | 24.13 ± 0.09 / **23.80 ± 0.20**    |
> >     | CFL [5]  | 23.69 ± 0.27 / **23.57 ± 0.20**       | 29.55 ± 0.38 / **29.29 ± 0.20**     | 24.33 ± 0.42 / **23.81 ± 0.15**    |
> >     | ADAFL [6]| 24.21 ± 0.23 / **23.55 ± 0.11**       | 34.03 ± 0.50 / **32.86 ± 0.36**     | 22.78 ± 0.27 / **22.48 ± 0.10**    |
> >
> > - If there are any specific additional methods you consider relevant, we will try our best to include them in our results.
> >
> > [3] Lin, Tsung-Yi et al. “Focal Loss for Dense Object Detection.” IEEE Transactions on Pattern Analysis and Machine Intelligence 42 (2017).
> >
> > [4] Baruch, Emanuel Ben et al. “Asymmetric Loss For Multi-Label Classification.” IEEE/CVF International Conference on Computer Vision (ICCV) (2020).
> >
> > [5] Smith, Leslie.  Cyclical Focal Loss. 10.48550/arXiv.2202.08978. (2022).
> >
> > [6] Arindam Ghosh, Thomas Schaaf, and Matt Gormley. AdaFocal: calibration-aware adaptive focal loss. In Proceedings of the 36th International Conference on Neural Information Processing Systems (NIPS '22).

---

> > > ### Author Response · Authors · 2024-11-24
> > > **Looking Forward to Your Thoughts**
> > >
> > > Dear Reviewer,
> > >
> > > We would like to thank you once again for your valuable feedback. We would greatly appreciate it if you could share your views on the revisions and the additional experiments we conducted to address your concerns and questions.

---

> > ### Comment · Reviewer_gKu8 · 2024-11-25
> > **more quetions**
> >
> > But as you claim " we propose a novel loss function, one-cold cross entropy (OCCE) loss", the core contribution of this paper is OCCE, which is exactly the same as RCE loss.

---

> > > ### Author Response · Authors · 2024-11-29
> > > **Response to Reviewer gKu8 (1/2)**
> > >
> > > We would like to sincerely thank you for your follow-up comment and for reiterating your concern regarding the novelty of the OCCE loss. While we acknowledge the apparent similarity between **a specific case** of OCCE—specifically, uniform OCCE—and RCE, we respectfully assert that OCCE is not "exactly the same as the RCE loss." To address this, **we present three key arguments and kindly request your feedback on these points.**
> > >
> > > 1. First, RCE in [1] is proposed as an alternative to CE, focusing narrowly on adversarial robustness. It suffers from inherent training instabilities and poor performance on large-scale datasets, as demonstrated in our experiments. In contrast, OCCE integrates class and anti-class modeling in a unified and balanced formulation (Equation 8), enabling it to overcome these limitations effectively. **Here are some results from our paper that clearly indicate that our approach outperforms RCE, and that RCE is not near to competitive as the classification task gets harder.**
> > >
> > >     ### **Closed Set Classification (Test Errors ↓)**
> > >     | **Dataset**       | **Loss**           | **ResNet18v2**         | **MobileNetv2**        | **DenseNet121**        |
> > >     |-------------------|--------------------|------------------------|------------------------|------------------------|
> > >     | **CIFAR-100**     | **CE (Baseline)**  | 24.98 ± 0.20           | 30.03 ± 0.35           | 25.06 ± 0.20           |
> > >     |                   | **RCE**            | 25.19 ± 0.23           | 50.93 ± 0.75           | 29.46 ± 0.12           |
> > >     |                   | **Proposed**       | **23.92 ± 0.23**       | **28.83 ± 0.25**       | **24.19 ± 0.12**       |
> > >     | **TinyImageNet**  | **CE (Baseline)**  | 36.80 ± 0.20           | 39.69 ± 0.49           | 38.71 ± 0.63           |
> > >     |                   | **RCE**            | 45.22 ± 0.24           | 73.34 ± 0.15           | 64.18 ± 0.48           |
> > >     |                   | **Proposed**       | **35.61 ± 0.13**       | **38.79 ± 0.22**       | **36.49 ± 0.52**       |
> > >
> > >     ### **Open Set Recognition (OSCR ↑)**
> > >     | **Method**   | **FashionMNIST**   | **CIFAR-100**   | **TinyImageNet**   |
> > >     |--------------|--------------------|-----------------|--------------------|
> > >     | **CE**       | 71.56             | 70.09          | 59.37             |
> > >     | **RCE**      | 70.62             | 64.47          | 37.50             |
> > >     | **Proposed**  | **73.93**         | **72.46**      | **61.94**         |
> > >
> > > 2. **Uniform OCCE is only an instantiation of OCCE.** OCCE can represent any desirable distribution for the complementary classes that sets the correct class to zero, hence the term "one cold." Even though we stated this flexibility of OCCE in the original manuscript, we have now conducted additional experiments to justify this empirically. We defined one-cold targets with non-uniform anti-class distributions that account for inter-class similarities. Specifically, we experimented with two variations:
> > >    - **Instance-based self-distillation**
> > >    - **Class-based self-distillation**
> > >
> > >     Both approaches utilize a soft one-cold encoding. The first accounts for instance-dependent similarities to each complementary class, while the second considers dataset-dependent similarities between classes. For implementation details, please refer to Section A.1: *Limitations of Uniform OCCE* (l.704).
> > >
> > >    | **Loss**                           | **MNIST**            | **CIFAR-10**        | **CIFAR-100**           | **TinyImageNet**        |
> > >    |------------------------------------|----------------------|---------------------|-------------------------|-------------------------|
> > >    | **CE (Baseline)**                  | 0.40 ± 0.04          | 5.45 ± 0.18         | 24.98 ± 0.40            | 36.80 ± 0.20            |
> > >    | **RCE**                            | 0.40 ± 0.05          | 5.55 ± 0.16         | 25.19 ± 0.23            | 45.22 ± 0.24            |
> > >    | **Proposed (Uniform OCCE)**        | 0.38 ± 0.01          | 5.40 ± 0.09         | 23.92 ± 0.23            | 35.61 ± 0.13            |
> > >    | **Proposed (Instance-based self-KD, a=0.1)** | 0.36 ± 0.02       | 5.38 ± 0.08         | **23.60 ± 0.32**        | 35.40 ± 0.10            |
> > >    | **Proposed (Class-based self-KD, a=0.5)**    | **0.33 ± 0.03**    | **5.28 ± 0.09**     | 23.71 ± 0.10            | **35.12 ± 0.16**        |
> > >
> > >    The best performance for each dataset is highlighted in bold. It is evident that the non-uniform OCCE targets give improvements on top of the uniform OCCE, which proves that our proposed approach effectively supports non-uniform target anti-class distributions—a concept that has not been proposed before. **Our core contribution lies in the combination of class and anti-class learning and how this can be effectively achieved in shared or decoupled layers, while remaining largely agnostic to the adopted anti-class distribution.**

---

> > > > ### Author Response · Authors · 2024-11-29
> > > > **Response to Reviewer gKu8 (2/2)**
> > > >
> > > > 3. Finally, regarding the uniform OCCE case, we provide new perspectives that are distinct from those in [1], as the motivations of the two papers are fundamentally different. We do not repeat prior work but instead add insights into why losses such as RCE work (e.g., through connections to neural collapse and independence deficit) and vice versa. While there is some novelty in these results, **they are not the core contribution of our paper**, as outlined in arguments 1 and 2.
> > > >
> > > >
> > > > These distinctions establish the originality and practical value of OCCE. We recognize that the initial manuscript's phrasing and limited discussion of these issues may have contributed to this misunderstanding. Your feedback has been invaluable in helping us revise the paper to clearly highlight the differences and novelty of OCCE.
> > > >
> > > > We hope that this clarification, along with the revisions made to the manuscript (clearly citing and discussing [1]), provides a better understanding of our contribution. **Our proposed approach using OCCE is clearly compared to RCE and outperforms RCE in all our experiments.** RCE has never been proposed for tackling tasks apart from adversarial robustness, and it also fails to handle large-scale datasets, as we demonstrate. The proposed OCCE formulation robustifies the ideas behind complementary class exploitation and provides a generic framework for achieving this effectively.
> > > >
> > > > **Please consider at least acknowledging our effort to fairly compare all the methods appearing in the relevant literature and giving our work a chance to be considered for publication. Therefore, we respectfully ask you to reconsider your evaluation in light of this clarification and share your thoughts on our three provided arguments.** If there are still specific aspects you would like us to clarify further, we would be more than happy to address them.
> > > >
> > > > [1] Pang, Tianyu et al. “Towards Robust Detection of Adversarial Examples.” Neural Information Processing Systems (2017).

---

> > > > > ### Author Response · Authors · 2024-12-01
> > > > > **Reminder: Rebuttal Discussion Period Closing Soon**
> > > > >
> > > > > Dear Reviewer,
> > > > >
> > > > > As the rebuttal discussion period is nearing its end, we kindly remind you of our response addressing your concerns about the contribution of our paper. We would greatly appreciate your feedback on the specific arguments we provided, along with your reasoning and justification for the score assigned to our paper, especially in light of our clarifications.
> > > > >
> > > > > Thank you for your time and consideration. We look forward to your thoughts.

---

### Official Review · Reviewer_619x · 2024-11-04

**Soundness:** 3
**Presentation:** 3
**Contribution:** 3
**Rating:** 8
**Confidence:** 4

**Summary:**

This paper proposes a new auxiliary loss function, called one-cold cross entropy (OCCE), for classification tasks. Unlike standard cross-entropy (CE), which relies on a one-hot vector representation, OCCE employs a one-cold vector, setting the true class to zero and all other classes to one. To apply this one-cold vector in OCCE, the logits of the final layer are inverted and passed through a standard softmax-cross-entropy loss. OCCE is then used as an auxiliary loss alongside CE. The proposed OCCE encourages a symmetric geometric structure among complementary classes, addressing the independence deficit problem and enhancing the degree of neural collapse.

Brief background on “independence deficit” and “neural collapse”: independence deficit is a phenomenon caused by the rank deficiency of deep neural networks, where classification confidence of a class can be linearly reproduced by the confidences of a small number of other (sometimes irrelevant) categories. And “neural collapse” signifies maximally separated and independent class representations.

The authors present experimental results in multiple settings for generalization in classification, open-set recognition, out of distribution recognition and transfer learning. In all reported results, OCCE improves the baseline.

**Strengths:**

+ Very well-written paper.
+ Tackles a fundamental problem and brings an interesting perspective.
+ Comprehensive set of experiments, where the proposed loss improves baseline performance.

**Weaknesses:**

- Although the experiments section is comprehensive, there is not a single result from literature (other papers) -- except the transfer learning in Table 2. Almost all experiments are comparisons to baseline. There is no comparison to SOTA. For example, the methods described in Related Work are competitors but there is no direct comparison with them, such as “baseline + OCCE” versus “baseline + competitor”. And, ideally, it would be more convincing if both the baseline and “baseline + competitor” results are taken from literature.
- Most key properties of OCCE were shown with a simple ResNet18v2 architecture. It would be more convincing to see more and larger encoders.
- In the abstract, robustness to “noise” is mentioned but I don’t see this in the experiments.

Minor:
Fig2 caption says z_0=3 but in the text you say 2.5.

**Questions:**

- “Relationships between non-target classes” -> this sounds ambiguous to me. Do you mean (i) the relationships between the target class and all complementary classes, or (ii) the relationships among all classes, or (iii) the relationships among complementary classes?
- Figure 1 can be made more clear. When you say “anti-class distribution for each point” and there are multiple points, do you superimpose them on top of each other and show just a single plot? This is not clear to me.
- Isn’t “anti-class” bad nomenclature? Target or non-target, they are all classes. I don’t know, It might be better to say “anti-ground truth” or “anti-true” or “anti-positive” instead of anti-class.
- How does squared difference (between y and y_hat) behave? It explicitly and symmetrically pulls down negative classes to zero and pushes the positive class to 1.
- How does OCCE do under class imbalance (long-tail)?
- In the tables, what is the value after plus minus? Standard deviation, standard error or variance?
- Does OCCE incur any kind of overhead during training?

Post-rebuttal edit: my questions above are sufficiently addressed by the authors.

---

> ### Author Response · Authors · 2024-11-20
> **Response to Reviewer 619x (1/2)**
>
> Thank you for your valuable feedback and time invested. We hope we will sufficiently address your concerns and questions and encourage a favorable reassessment of our work.
>
> **(W1)**: Although the experiments section is comprehensive, there is not a single result from literature (other papers) -- except the transfer learning in Table 2. Almost all experiments are comparisons to baseline. There is no comparison to SOTA. For example, the methods described in Related Work are competitors but there is no direct comparison with them, such as “baseline + OCCE” versus “baseline + competitor”. And, ideally, it would be more convincing if both the baseline and “baseline + competitor” results are taken from literature.
>
> - We have conducted additional experiments using 4 additional methods ([1], [2], [3], [4]) proposed in the literature, that were highlighted by other reviewers leading to a total of 9 competitors which we compare against. Table 1 presents a direct comparison between these methods, showing their performance with and without the proposed approach. This enables us to evaluate the performance of the methods both in isolation, as well as when combined with the proposed approach, covering this way the comparisons asked by the reviewer, while also providing additional results. We demonstrate that in all cases using the proposed method improves the results. To improve space utilization in Table 1, we used the same cell to present both results separated by a slash, in the format (competitor / proposed).
> - In our experiments we use official implementations and hyperparameters from the referenced works, ensuring fair comparisons under consistent training conditions. Full training details and code are provided to ensure reproducibility.
> - At present, we have completed these comparisons for CIFAR-100, while the results for TinyImageNet will be finalized and included in the updated version of the paper. The results for 4 additional methods (and baseline cross entropy) for CIFAR-100 are:
>
>     |Test Error (↓)| ResNet18v2 (Competitor / +OCCE)         | MobileNetv2 (Competitor / +OCCE)      | DenseNet121 (Competitor / +OCCE)     |
>     |-------|---------------------------------------|-------------------------------------|------------------------------------|
>     | CE (Baseline)    | 24.98 ± 0.20 / **23.92 ± 0.23**       | 30.03 ± 0.35 / **28.83 ± 0.25**     | 25.06 ± 0.20 / **24.19 ± 0.12**    |
>     | FL [1]  | 25.60 ± 0.14 / **23.60 ± 0.45**       | 30.98 ± 0.20 / **29.45 ± 0.23**     | 26.13 ± 0.05 / **24.16 ± 0.19**    |
>     | ASL [2]  | 24.81 ± 0.11 / **23.75 ± 0.18**       | 29.76 ± 0.06 / **29.10 ± 0.33**     | 24.13 ± 0.09 / **23.80 ± 0.20**    |
>     | CFL [3]  | 23.69 ± 0.27 / **23.57 ± 0.20**       | 29.55 ± 0.38 / **29.29 ± 0.20**     | 24.33 ± 0.42 / **23.81 ± 0.15**    |
>     | ADAFL [4]| 24.21 ± 0.23 / **23.55 ± 0.11**       | 34.03 ± 0.50 / **32.86 ± 0.36**     | 22.78 ± 0.27 / **22.48 ± 0.10**    |
>
> - The results for the other 4 competitors are provided in Table 1 of the paper and are repeated below (only for CIFAR-100), confirming that using the proposed approach always leads to improvements.
>
>     | Test Error (↓) | ResNet18v2 (Competitor / +OCCE)         | MobileNetv2 (Competitor / +OCCE)      | DenseNet121 (Competitor / +OCCE)     |
>     |----------------|---------------------------------------|-------------------------------------|------------------------------------|
>     | LSCE           | 23.26 ± 0.14 / **22.97 ± 0.16**       | 29.26 ± 0.16 / **29.14 ± 0.30**     | 23.59 ± 0.21 / **23.43 ± 0.06**    |
>     | CE+NL          | 25.10 ± 0.14 / **23.90 ± 0.33**       | 29.95 ± 0.22 / **28.89 ± 0.37**     | 25.48 ± 0.81 / **24.71 ± 0.44**    |
>     | COT            | 24.52 ± 0.19 / **23.21 ± 0.26**       | 29.57 ± 0.11 / **28.52 ± 0.32**     | 23.70 ± 0.75 / **22.27 ± 0.26**    |
>     | CCE            | 25.36 ± 0.29 / **23.57 ± 0.22**       | 29.81 ± 0.25 / **28.72 ± 0.31**     | 25.29 ± 0.34 / **23.80 ± 0.38**    |
>
> **(W2)**: Most key properties of OCCE were shown with a simple ResNet18v2 architecture. It would be more convincing to see more and larger encoders.
>
> - To demonstrate the generality of the proposed approach we have already evaluated it using a larger Swin-T vision transformer. The results are presented in Appendix (A.5.2), showing patterns highly consistent with the observations on resnets, which confirms that the behavior of our approach is agnostic to the employed architecture.
>
> [1] Lin, Tsung-Yi et al. “Focal Loss for Dense Object Detection.” IEEE Transactions on Pattern Analysis and Machine Intelligence (2017).
>
> [2] Baruch, Emanuel Ben et al. “Asymmetric Loss For Multi-Label Classification.” IEEE/CVF International Conference on Computer Vision (ICCV) (2020).
>
> [3] Smith, Leslie.  Cyclical Focal Loss. 10.48550/arXiv.2202.08978. (2022).
>
> [4] Arindam Ghosh, Thomas Schaaf, and Matt Gormley. AdaFocal: calibration-aware adaptive focal loss.  (NIPS '22).

---

> > ### Author Response · Authors · 2024-11-20
> > **Response to Reviewer 619x (2/2)**
> >
> > **(W3)**: In the abstract, robustness to “noise” is mentioned but I don’t see this in the experiments.
> >
> > - The sentence in question is:
> >
> >   *"Specifically, for each class, we define an anti-class, which consists of everything that is not part of the target class—this includes all complementary classes as well as out-of-distribution samples, noise, or in general any instance that does not belong to the true class."*
> >
> > - The term "noise" was used to emphasize that an anti-class encompasses the entire complementary subspace of a class, including instances such as noise, rather than referring to a specific class. We acknowledge that this phrasing may cause confusion and clarify that we are **not** claiming robustness to noise. To avoid misunderstanding, we will appropriately revise the abstract and remove the term "noise."
> >
> > **(Q1)**: “Relationships between non-target classes” this sounds ambiguous to me. Do you mean (i) the relationships between the target class and all complementary classes, or (ii) the relationships among all classes, or (iii) the relationships among complementary classes?
> >
> > - We are referring to the relationships among complementary classes. Throughout the paper, the terms "non-target classes" and "complementary classes" are used interchangeably.
> >
> > **(Q2)**: Figure 1 can be made more clear. When you say “anti-class distribution for each point” and there are multiple points, do you superimpose them on top of each other and show just a single plot? This is not clear to me.
> >
> > - Figure 1 represents a core concept of our method, but we acknowledge it could be explained more clearly. Each point in the grid corresponds to a specific $(x_1, x_2)$ coordinate in the input space $X$, (not limited to visible class samples). Each point is passed through a trained MLP, its logits are negated and passed through a softmax to provide an anti-class distribution (membership of that specific point of the grid to each anti-class). The color assigned to each point is calculated with the linear combination between the anti-class distribution of that point and the colors of the classes. The key insight is that, under cross entropy, dominant anti-classes emerge in different regions, while under uniform OCCE, the anti-class distribution becomes uniform (indicated by a more neutral color).
> > - We will refine the Figure's 1 caption to improve clarity following the Reviewer's suggestion.
> >
> > **(Q3)**: Isn’t “anti-class” bad nomenclature? Target or non-target, they are all classes. I don’t know, It might be better to say “anti-ground truth” or “anti-true” or “anti-positive” instead of anti-class.
> >
> > - The term “anti-class” was chosen to emphasize that it represents the entire complementary space of a class, not just specific non-target classes. While alternative terms like “anti-ground truth” or “anti-positive” might also be appropriate, we felt they might direct the reader towards contrastive learning or negative learning perspectives, which differ from our intention.
> >
> > **(Q4)**: How does squared difference (between y and y\_hat) behave? It explicitly and symmetrically pulls down negative classes to zero and pushes the positive class to 1.
> >
> > - MSE does indeed symmetrically reduce the probabilities of incorrect classes to 0 and increases the correct class to 1. However, its linear gradients can lead to slower convergence compared to cross entropy, which emphasizes correcting large errors more strongly through its logarithmic gradients. OCCE, in contrast, introduces a target anti-class distribution for complementary classes, providing stronger training signals for non-target classes (illustrated in Figure 2), which both cross entropy and squared differences do not.
> >
> > **(Q5, Q6, Q7)**: How does OCCE do under class imbalance (long-tail)? In the tables, what is the value after plus minus? Standard deviation, standard error or variance? Does OCCE incur any kind of overhead during training?
> >
> > - We conducted additional experiments to evaluate OCCE under exponential imbalance. We trained our models on CIFAR-100 with factors 0.1, 0.05, 0.02, and 0.01. OCCE improves performance across all cases. Under extremely imbalanced scenarios the gains reduce, yet still perform significantly better than the baseline.
> > - The plus/minus after the value shows the standard deviation between different runs.
> > - OCCE does not incur any overhead due to a single forward and backward pass needed to compute both CE and OCCE.
> >
> >     | Test Error (↓)        | Imb Factor 0.1      | Imb Factor 0.05     | Imb Factor 0.02     | Imb Factor 0.01     |
> >     |------------------------|---------------------|---------------------|---------------------|---------------------|
> >     | CE (Baseline)          | 40.17 ± 0.26       | 45.26 ± 0.16       | 53.17 ± 0.45       | 58.11 ± 0.36       |
> >     | CE + OCCE (γ=1)        | **38.90 ± 0.44**   | **44.50 ± 0.41**   | **52.03 ± 0.37**   | **57.59 ± 0.08**   |

---

> > > ### Comment · Reviewer_619x · 2024-11-23
> > >
> > > I thank the authors for thoroughly addressing my concerns and questions.

---

### Author Response · Authors · 2024-11-20
**General Response to Reviewers**

Dear reviewers,

We thank you all for your valuable time, comments, and the opportunity you have given us to improve our paper.

We have responded to each reviewer individually to address any comments. We want to provide a brief summary:

1. We conducted experiments on ImageNet with a standard ResNet-50, showing that our proposed approach leads to 2.45% and 4.30% relative improvements in test errors (0.81% and 0.35% in accuracy terms), compared to cross entropy.
2. We enriched Table 1 of the paper by comparing against several relevant losses suggested by the reviewers, showing that combining OCCE with all of these losses improves their performance consistently, with an average improvement of 3.5% on test errors on CIFAR-100 (similar to our previous findings).
3. We compared directly against SupCon [1] (as asked by Reviewer Q6sr and Reviewer auHK), showing superior performance both in terms of accuracy and training time in a given configuration, while also discussing the fundamental differences between the two approaches.
4. Reviewer gKu8 highlighted a relevant work [2]. The loss proposed in [2] is only a specific instantiation of our proposed framework, which focuses on providing adversarial robustness. We propose a much broader idea and methodology under a completely different motivation, which can effectively handle a wider range of learning scenarios, as demonstrated in our paper. Additionally, in our paper we demonstrate that the loss proposed in [2] cannot be used for general classification tasks (see Table 5), which shows a significant drop in performance as the scale of the dataset increases. On the other hand, the proposed method provides a more general framework with more flexibility (e.g. using a shared or decoupled layer for the anti-classes or potentially using different target anti-class distributions). This always leads to improved performance compared to [2]. A more in-depth discussion is provided in our response to Reviewer gKu8.
5. We addressed points of confusion raised by the reviewers and explained how we revise the paper to improve clarity and presentation.


[1] Khosla, P., Teterwak, P., Wang, C., Sarna, A., Tian, Y., Isola, P., Maschinot, A., Liu, C., & Krishnan, D.  Supervised contrastive learning. Advances in Neural Information Processing Systems (2020).

[2] Pang, Tianyu et al. “Towards Robust Detection of Adversarial Examples.” Neural Information Processing Systems (2017).

---

### Meta-Review · Area_Chair_MBF4 · 2024-12-25

**Metareview:**

This paper was reviewed by four experts in the field. The paper received mixed review ratings of 8, 3, 5, and 6. The AC read the paper, reviews, and rebuttal carefully. This is a borderline paper. This paper has merit by proposing an interesting idea and providing some theoretical justification. The major issue of the paper, as also pointed out by some reviewers, is the lack of strong empirical results, especially results beyond small datasets like CIFAR and TinyImageNet. It is unclear whether the loss can be helpful or complimentary to the existing paradigm used for real-world problems. The ACs believed that the concerns around the empirical results outweighed the paper's strengths. The AC would suggest the authors show more empirical results and larger scale datasets to justify the proposed loss better. The authors are encouraged to consider the reviewers' comments when revising and resubmitting the paper.

**Additional Comments On Reviewer Discussion:**

Reviewers' ratings are mixed after the rebuttal. Reviewer 619x and auHK are most positive about the paper, but they all have concerns about the empirical results of the paper. This is a borderline paper, but the AC believed large-scale experiments are needed to justify the real-world implications of the proposed loss.

---

### Decision · Program_Chairs · 2025-01-22

Reject